# A cdk1 gradient guides surface contraction waves in oocytes

Johanna Bischof [1], Christoph A. Brand[2], Kálmán Somogyi[1], Imre Májer[1], Sarah Thome[1], Masashi Mori[1], Ulrich S. Schwarz [2] & Péter Lénárt [1]

Surface contraction waves (SCWs) in oocytes and embryos lead to large-scale shape changes coupled to cell cycle transitions and are spatially coordinated with the cell axis. Here, we show that SCWs in the starfish oocyte are generated by a traveling band of myosin II-driven cortical contractility. At the front of the band, contractility is activated by removal of cdk1 inhibition of the RhoA/RhoA kinase/myosin II signaling module, while at the rear, contractility is switched off by negative feedback originating downstream of RhoA kinase. The SCW's directionality and speed are controlled by a spatiotemporal gradient of cdk1-cyclinB. This gradient is formed by the release of cdk1-cyclinB from the asymmetrically located nucleus, and progressive degradation of cyclinB. By combining quantitative imaging, biochemical and mechanical perturbations with mathematical modeling, we demonstrate that the SCWs result from the spatiotemporal integration of two conserved regulatory modules, cdk1-cyclinB for cell cycle regulation and RhoA/Rok/NMYII for actomyosin contractility.

[1] Cell Biology and Biophysics Unit, European Molecular Biology Laboratory (EMBL), Meyerhofstrasse 1, 69117 Heidelberg, Germany. [2] Institute for Theoretical Physics and BioQuant, Heidelberg University, Philosophenweg 19, 69120 Heidelberg, Germany. Correspondence and requests for materials should be addressed to P.L. (email: peter.lenart@embl.de)

I t is one of the central questions in cell biology how the cell's biochemistry is translated into force and shape changes, and how, in turn, physical factors impact cellular processes such as cell division, migration, and development[1, 2]. Here, we use surface contraction waves (SCWs), which are conserved, stereotypical shape changes in dividing oocytes and embryos, as a model system to reveal such principles.

SCWs provide spatial coordination for the divisions of large egg cells[3], and have been suggested to support the asymmetric meiotic divisions in oocytes[4], as well as to localize embryonic patterning determinants in the egg cytoplasm[5–9]. SCWs were first described in axolotl eggs[10] and have been investigated in oocytes and eggs of *Xenopus*[6, 11], starfish[4], and jellyfish[5], and similar contractile behaviors have been observed in many animal species[3, 12–16]. In the rapid divisions of the millimeter-sized amphibian oocytes, two SCWs are observed simultaneously on different parts of the egg surface correlated with cell cycle transitions: SCWa is a relaxation wave coinciding with entry into M-phase, followed by SCWb, a contraction wave at exit from M-phase[17–20]. This general behavior seems to be conserved in species with an intermediate oocyte size of hundreds of micrometers, such as starfish or jellyfish, where SCWa and SCWb are similarly present, although they are separated in time and pass across the cell individually[4].

Despite their conservation and functional importance, the molecular mechanisms that drive the SCWs and link them to the cell cycle remained to be explored. Here, we studied SCWs in the powerful model system of starfish oocyte meiosis[4, 21], which is well suited for quantitative live imaging combined with physical and biochemical perturbations. We show that the shape changes during the SCW arise from a transient band of cortical actomyosin contraction traveling across the oocyte. This contraction is switched on by releasing a block of the conserved RhoA/RhoA kinase signaling to non-muscle myosin II by the cell cycle kinase cdk1, ensuring temporal coupling to M-phase progression. We furthermore show that the spatial coordination of the directionality of the SCW with the cell axis is determined by a cytoplasmic gradient of cdk1-cyclinB, set by the release of nuclear cdk1-cyclinB at the beginning of meiosis and driven across the cell by cyclinB degradation. We present a comprehensive quantitative model for how the spatiotemporal coupling of two conserved molecular modules coordinates changes in cell shape in large cells.

## Results

**The SCW is a traveling band of cortical contraction.** There are four distinct waves of shape change during meiosis in the starfish oocyte: one occurring at entry into meiosis I (SCWa I in the *Xenopus* nomenclature), one at exit from meiosis I (SCWb I), which then reoccur in the second meiotic division (SCWa II and SCWb II) (Fig. 1a). Here, we focused on the most prominent of these shape changes, the contraction wave at the end of meiosis I (referred to simply as SCW hereafter).

To quantify the shape changes occurring during SCWs we segmented the cell outline and calculated the curvature radii along the perimeter (Supplementary Fig. 1a). As revealed by these quantifications, the SCW arises from a flattening of the cortex starting at the vegetal pole (V) opposite of the spindle apparatus. This flattening then moves across the cell as a band and arrives at the animal pole (A) coinciding with polar body formation (Fig. 1b, Supplementary Movie 1). The dynamics of the shape change can be visualized by mapping values of curvature radii to a line and plotting these lines for all time points, generating a kymograph (Fig. 1d). As evident from the kymographs, the SCW has a duration of $7 \pm 1.5$ min, during which it moves across the

oocyte with a speed of $42.4 \pm 7.9\,\mu m/min$ ($n = 55$) causing the strongest deformation near the equator of the oocyte.

To gain an understanding of the underlying mechanics, we simulated these shape changes using a three-dimensional (3D) model of the cell surface approximated by a contractile triangular mesh (for details see Methods, Supplementary Table 1). This simulation revealed that locally contracting the surface by increasing the active tension in a band perpendicular to the A–V axis, and moving this band at a constant speed across the cell results in shape changes that qualitatively reproduce the SCW, including the peak change in curvature occurring near the equator (Fig. 1c, d). Simulations of other possible scenarios, such as a wave front of increased contractility moving across the oocyte, did not reproduce the experimental observations (Supplementary Fig. 1b). The traveling band model predicts that 'softening' the oocyte, i.e., reducing its elastic modulus, should result in larger deformations (Fig. 1e). To test this, we removed the jelly layer surrounding the oocytes either by acidic sea water[22] or enzymatic treatment[23] which reduced the effective Laplace tension by ~50% as measured by micropipette suction[24] (Supplementary Fig. 1c, d). Indeed, SCWs caused much larger deformations after jelly removal (Fig. 1f, g, Supplementary Fig. 1e), supporting the traveling band model.

**Contraction is mediated by the RhoA/Rok/Myosin II module.** To identify the molecular drivers of contractility, we tested the localization of non-muscle myosin II (NMYII). Fluorescently tagged NMYII heavy chain (NMYIIhc-mEGFP) was recruited to the contracting cortex during the SCW (Fig. 2a, b and Supplementary Movie 2). Quantification of the fluorescence in the cytoplasmic area underlying the cortex revealed a concomitant decrease (~20%) compared with fluorescence intensity in metaphase, indicating that NMYII is recruited to the contractile cortex locally from the underlying cytoplasm (Fig. 2b). To confirm that NMYII activity is required, we treated oocytes with blebbistatin, which almost completely prevented contraction (Fig. 2d) while over-activation of NMYII by overexpressing myosin regulatory light chain (MRLC) resulted in stronger contraction (Fig. 2d). Depolymerization or stabilization of actin filaments by cytochalasin D or phalloidin, respectively, also blocked contraction (Fig. 2d), indicating that the SCW is generated by NMYII-driven actomyosin contractility.

To map the pathway leading to cortical recruitment and activation of NMYII, we tested the involvement of the conserved GTPase RhoA using a fluorescent reporter for active RhoA (EGFP-rGBD[25]). Note that in starfish transcriptomes we identified only a single Rho mRNA encoding a single Rho protein to which we refer here as RhoA. This marker for active RhoA showed a strong and transient RhoA activity coinciding with the traveling band of contractility during the SCW (Fig. 2g, Supplementary Movie 3). RhoA activity was specifically required for actomyosin contraction, as RhoA or Rho-kinase (Rok, again only a single Rok is present in starfish) inhibition with C3 exoenzyme or Y-27632, respectively, blocked the SCW (Fig. 2e, f). By contrast, we observed no effect on the strength of the SCW after treatment with ML-7, commonly used to inhibit myosin light chain kinase (Fig. 2f). These data indicate that transient local activation of RhoA drives the SCW through Rok mediated activation of NMYII (Fig. 2c).

Interestingly, when contraction was inhibited by the Rok inhibitor Y-27632, RhoA activity still progressed across the oocyte, but it did so as a wave front rather than as a band (Fig. 2h, Supplementary Movie 4). The width of the traveling band in the untreated case therefore must be set by inactivation of RhoA that depends on Rok activity itself or signaling downstream of Rok,

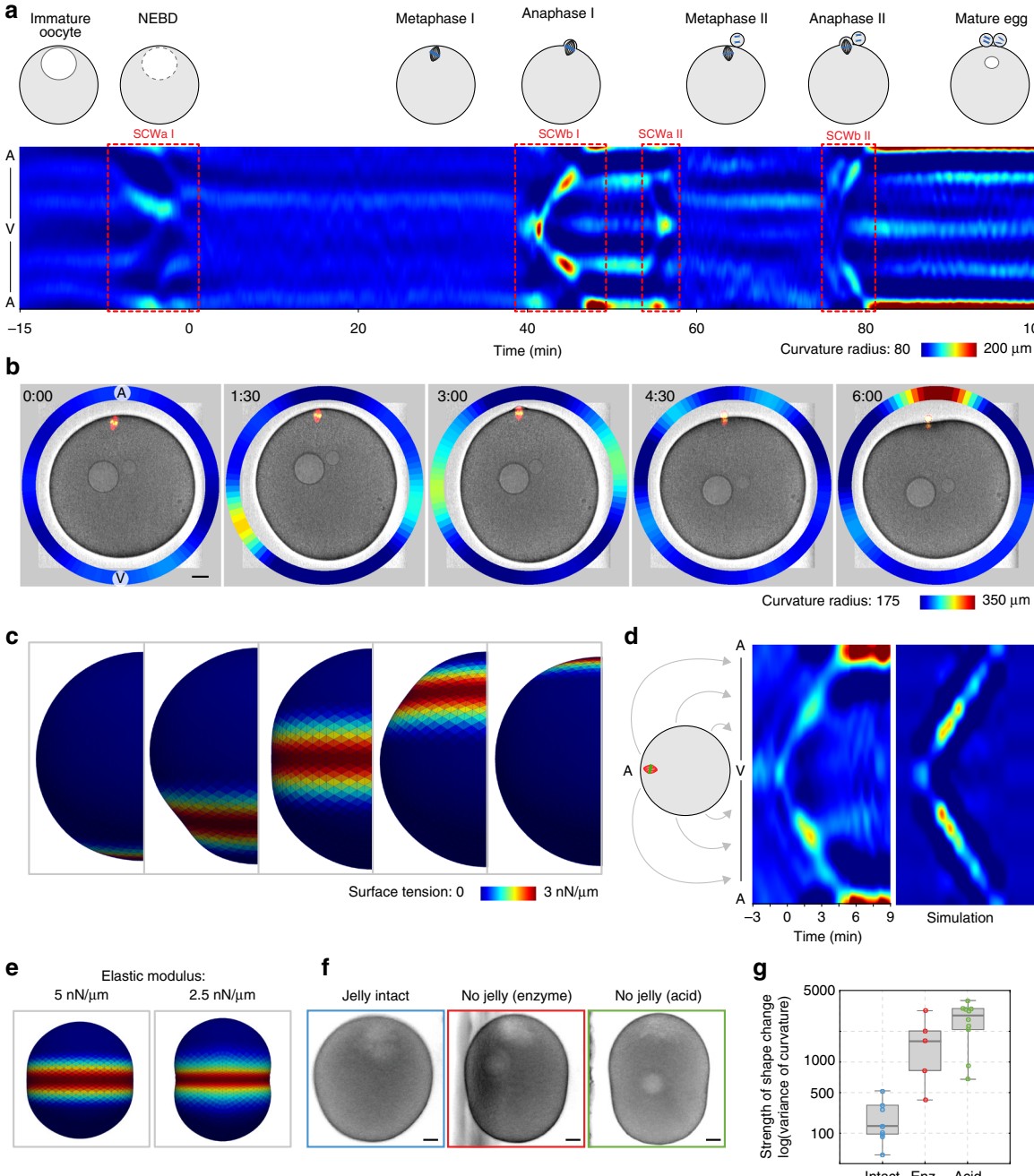

**Fig. 1** The SCW is a band of cortical contraction traveling across the oocyte. **a** Kymograph of radii of curvature during the complete meiosis in starfish oocytes, along with schematic representations of key stages of the divisions. **b** Overview of the SCW in starfish oocytes. Frames selected from a transmitted light movie across the equatorial plane along the animal–vegetal (A–V) axis of an oocyte with fluorescence channels overlaid labeling microtubules (red, EB3-mEGFP3) and chromosomes (green, mCherry-H2B). For the full movie see Supplementary Movie 1. The ring plot shows the radii of curvature of the cortex encoded in the pseudo-color scale shown on the bottom right. Scale bar=20 μm; time is given as m:ss relative to beginning of SCW. **c** Rendering of the 3D mechanical model of the oocyte during the SCW. Surface tension is encoded using the pseudo-color scale shown on the bottom right. **d** Kymographs of cortical curvature radii during the SCW of the oocyte shown in **b**, and of the simulation shown in **c**. Kymographs show curvature radii along the perimeter of the oocyte on the y-axis as shown on the schematics on the left, x-axis is time. Radii of curvature values are encoded in the pseudo-color scale as in **b**. Kymograph of the simulation was generated by calculating curvature of individual steady state outcomes of simulation of the progress of the contraction. **e** Frames selected at the maximal deformation during the SCW of simulations with high and low elastic moduli. **f** Frames of oocytes with either intact jelly or jelly layer removed (by enzymatic or acidic sea water treatment) as the SCW passes the equator. Scale bar=20 μm. **g** Comparison of the strength of the shape change during SCW as measured by the variance of curvature radii during the SCW in oocytes with and without jelly, jelly removed by the indicated method. Dot plots of measurements of individual oocytes overlaid with box plots of the same data

implying a negative feedback loop originating downstream of Rok. A similar negative feedback in the RhoA signaling module has recently been demonstrated in cytokinesis[26].

In summary, our data show that a spatiotemporally coordinated, transient activation of the conserved RhoA signaling module drives SCWs, by generating a traveling band of

actomyosin contractility that sweeps across the oocyte surface, the width of which is set by negative feedback internal to the RhoA signaling module (Fig. 2c).

Interestingly, manipulating the SCWs, either by increasing the strength of contraction by MRLC overexpression or suppressing it by blebbistatin, had no effect on the size of polar body

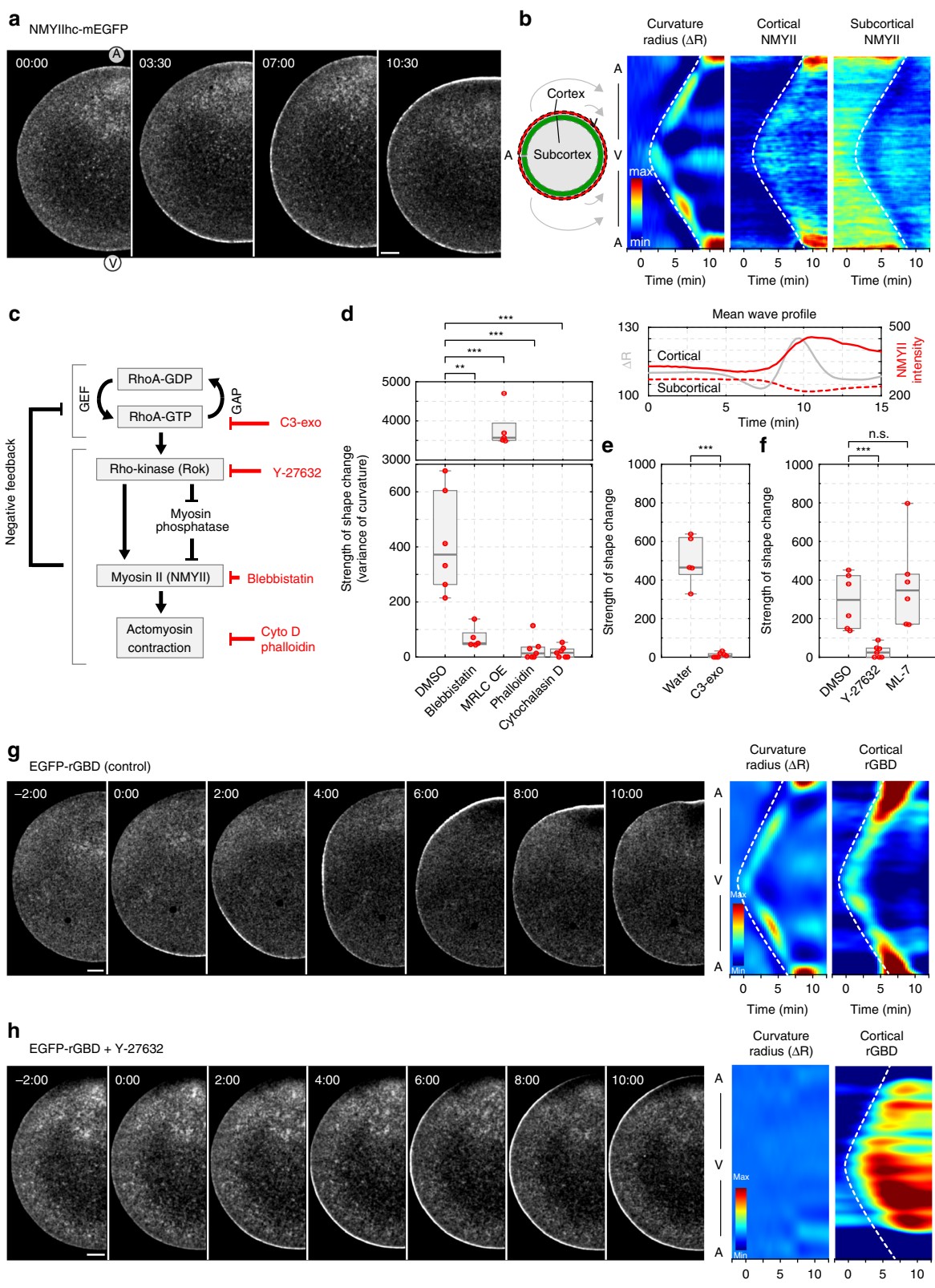

protrusion (Supplementary Fig. 2a, b). This indicates that protrusion of the polar body is independent of SCWs and is driven by a distinct mechanism. However, the closing off of the polar body does use the same molecular components (Supplementary Fig. 2c), and therefore, while the above treatments affecting RhoA-activated actomyosin contractility did not affect protrusion, they did prevent 'pinching off' the polar body, leading to a re-adsorption of the polar body in inhibitor treated cases (Supplementary Fig. 2b).

**RhoA is activated by release from cdk1 inhibition**. Having clarified the mechanism of contraction, we next asked how this traveling band of contractility is initiated and guided in space and how its timing is coordinated with the cell cycle. It is known from mitosis that RhoA activity can be controlled by inhibition of its guanine nucleotide exchange factor (GEF) Ect2 through phosphorylation by cdk1-cyclinB[27]. Our data are consistent with Ect2 being the GEF responsible for controlling RhoA for SCWs (Supplementary Fig. 3a, b), although due to the lack of tools to deplete the endogenous protein in oocytes we could at present not fully exclude alternative mechanisms.

To test if changing cdk1 activity can trigger RhoA activation, we inhibited cdk1 with RO-3306. Applying the inhibitor to the whole oocyte resulted in global recruitment of active RhoA to the cortex indicating that RhoA is indeed normally kept inactive by cdk1 activity (Fig. 3b), consistent with previous studies[26]. If cdk1 inhibition is the major trigger of the SCW, we should be able to change the direction of the SCW by acute local inhibition of the kinase. Indeed, local application of RO-3306 or Roscovitine caused localized recruitment of RhoA and subsequent spreading of RhoA activity from this point across the cell, allowing a complete reversal of the direction of the SCW when applied to the animal pole (Fig. 3a, c, Supplementary Fig. 4a). Depolymerization of microtubules by nocodazole had no effect on the strength or overall pattern of the SCWs, showing that the cdk1 regulation is independent of the small meiotic spindle at the animal pole (Supplementary Fig. 4b, c).

These data show that recruitment and activation of RhoA is controlled by inactivation of cdk1. Inactivation of cdk1 occurs at meiotic exit and thereby ensures synchronization of the SCW with cell cycle progression. Interestingly, this data furthermore suggests that cdk1 activity may not only temporally regulate the SCW but also control its spatial pattern.

**cdk1 forms a spatiotemporal gradient matching the SCW**. To test if the spatial pattern of cdk1 controls the SCW, we visualized cdk1 using a fluorescently tagged version of its regulatory subunit cyclinB (cyclinB-EGFP) as a well-established reporter[28, 29]. As previously reported[30], cyclinB-EGFP is imported into the nucleus in prophase, which concentrates cdk1-cyclinB at the animal pole at the beginning of meiosis I (Fig. 3d, Supplementary Movie 5).

After nuclear envelope breakdown (NEBD) this accumulation leads to the formation of a gradient along the animal-vegetal axis, which is maintained until the end of metaphase I, when it collapses concomitantly with the SCW passing across the cell (Fig. 3d).

The shape of the gradient evolves as meiosis progresses (Fig. 3e, i, j) since cdk1-cyclinB diffuses slowly in the large oocyte and cyclinB is gradually degraded. Strikingly, overlaying the temporal change in cyclinB-EGFP intensity with the change in surface curvature revealed a near perfect overlap between the cdk1-cyclinB profile and the SCW (Fig. 3e). This close spatiotemporal correlation suggests that the contractile band that constitutes the SCW is triggered by RhoA activation below a critical cdk1 activity threshold, and then follows this threshold along the vegetal to animal axis as cyclinB is degraded.

To validate that cyclinB degradation can indeed shape the observed cdk1-cyclinB gradient, we developed a reaction-diffusion model in the 3D geometry of the starfish oocyte, that incorporates realistic biophysical parameters[31], a minimal set of reactions that drive cyclinB degradation (by APC/C, activated in a delayed positive feedback loop by cdk1-cyclinB[32, 33]), and the nuclear concentration of cdk1-cyclinB determined via fluorescence imaging as initial conditions (Fig. 3g, h, for details see Methods and Supplementary Table 2). Using this minimal set of assumptions, we were able to reproduce the shape of the experimentally observed cdk1-cyclinB gradient (Fig. 3f, k, l), validating the hypothesis that nuclear accumulation plus diffusion of cyclinB is sufficient to set the directionality of the gradient, even if in practice the underlying biochemical reactions are likely to be more complex.

**The SCW front is guided by the cdk1-cyclinB gradient**. If the cdk1-cyclinB gradient guides the SCW front by locally activating RhoA wherever the critical lower threshold of cdk1 activity is reached, changing the directionality of the gradient should redirect the SCW. To test this, we first relocated the nucleus by gentle centrifugation away from the cortically anchored centrosomes, which mark the original animal pole[34]. As predicted, in these oocytes the direction of the SCWs invariably followed the new animal-vegetal axis set up by the new nuclear position (Fig. 4a) and no longer correlated with the original animal-vegetal axis. Second, we moved different parts of the cortex to the furthest distance from the nucleus by forcing oocytes into a variety of shapes in microfabricated chambers[35]. RhoA accumulation approximately started at the site furthest from the animal pole, and initiated at multiple sites if they were equidistant (Fig. 4b). Consistent with the above local cdk1 inhibition experiments, this validates the prediction that it is low cdk1 activity that sets the site where the SCW initiates independent of the animal-vegetal axis. In addition, we observed that the animal-vegetal distance and the speed of the SCW in oocytes with a

**Fig. 2** The SCW is mediated by the RhoA/Rok/NMYII module. **a** Selected frames of a time-lapse recording of an oocyte expressing the heavy chain of non-muscle myosin II (NMYIIhc)-mEGFP during SCW. Single confocal slice along the equatorial plane across the A–V axis; scale bar 20 μm; time is in mm:ss. For the complete movie see Supplementary Movie 2. **b** Kymographs showing radii of curvature as in Fig. 1c, and the fluorescence signal of NMYIIhc-EGFP measured at the cortex and in a subcortical region simultaneously over time, as illustrated by the scheme on the left. Underneath, fluorescence intensity plot of the cortical (solid line) and subcortical NMYIIhc-EGFP (dashed line) signal, compared to curvature change (gray line) averaged along the wave front. **c** Scheme illustrating the signaling pathway controlling SCWs in starfish oocytes and the inhibitors used in this study to interfere with the respective components. **d–f** Quantification of the strength of the shape change during the SCW for oocytes treated with inhibitors as indicated. Dot plots of measurements of individual oocytes overlaid with box plots of the same data. ***$p < 0.001$, **$p < 0.01$, n.s. not significant, determined via ANOVA. **g** Selected frames of a time-lapse recording of an oocyte expressing RhoA-GTP reporter (EGFP-rGBD) during the SCW and corresponding kymographs of curvature radii and cortical fluorescence intensities as in **b**. Single confocal slice along the equatorial plane across the A–V axis is shown; scale bar=20 μm, time is in mm:ss. For the complete movie see Supplementary Movie 3. **h** Oocyte expressing EGFP-rGBD and imaged as in **g**, injected with the Rok inhibitor Y-27632, selected frames and kymographs of radii of curvature and cortical fluorescence intensities are shown. Scale bar=20 μm; time is in mm:ss. For the complete movie see Supplementary Movie 4

variety of shapes show a strong correlation (Fig. 4d), indicating that the gradient is scalable and changes its slope as it is compressed or expanded in the spatial dimension.

To change the direction of the cdk1 activity gradient more specifically than by moving nucleus and cortex relative to each other, we microinjected purified active cdk1-cyclinB protein into the vegetal hemisphere. This creates a pattern of high cdk1-cyclinB concentration at both poles, which should set the site of SCW initiation near the oocyte equator, where cdk1-cyclinB concentration is lowest. Indeed, we observed the initial

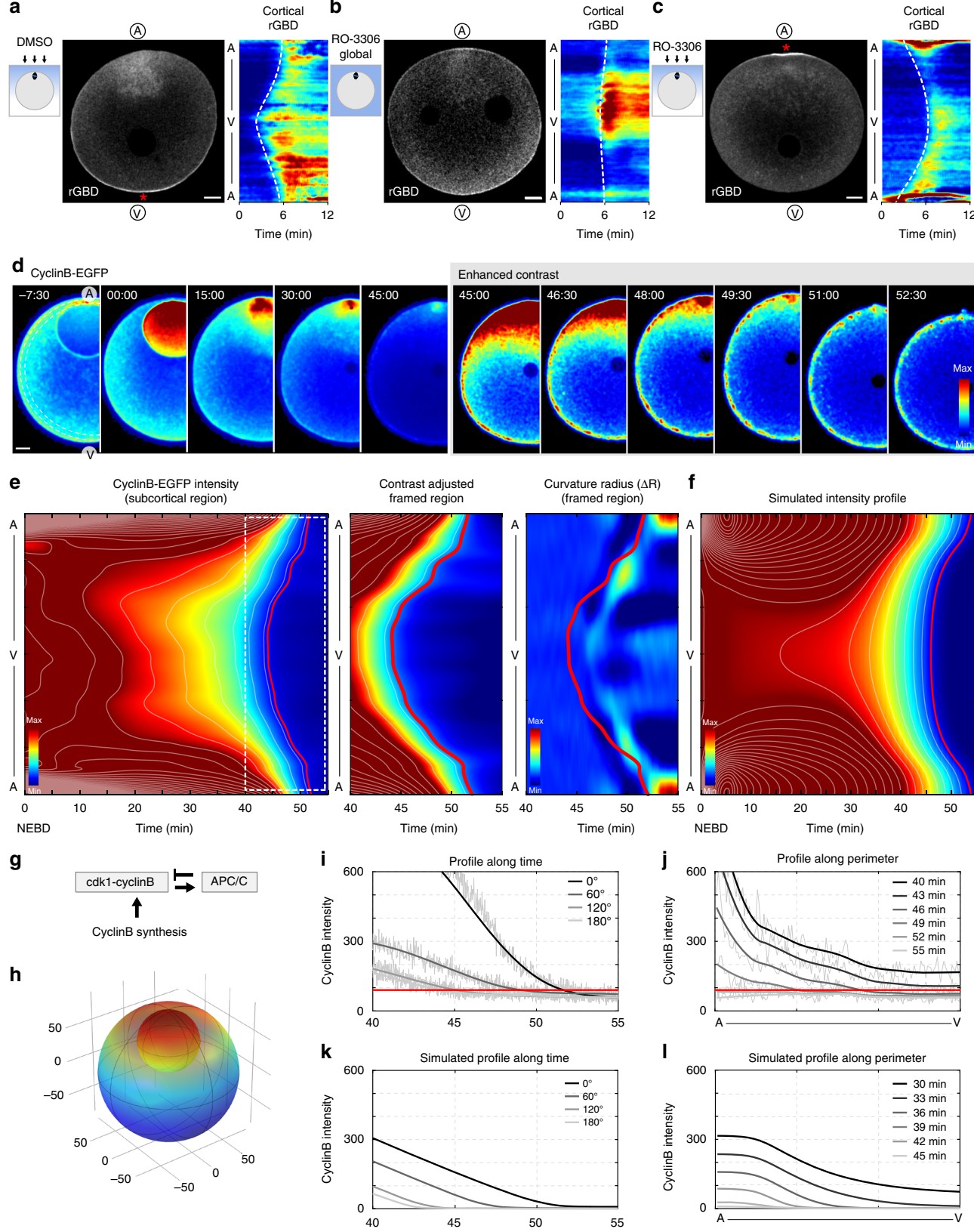

recruitment of active RhoA near the equator between the nuclear region and the injection site (Fig. 4c). Consistently, we also observed that overexpressing cyclinB-EGFP in oocytes lead to a flattening of the gradient in the vegetal hemisphere, which was tracked by the SCW (Supplementary Fig. 5a).

These perturbation experiments confirm our hypothesis that a cdk1-cyclinB gradient originating from the nucleus directs the SCW, which initiates at the location where the critical lower threshold of cdk1 activity is reached first and then moves along the gradient as this lower threshold progresses across the cell by degradation of cyclinB.

## Discussion

We present here a comprehensive model for how changes at the molecular level pattern the cell-scale behavior of SCWs. We show that the SCW is driven by the highly conserved RhoA/Rok/NMYII signaling module. The wave front is patterned by a spatiotemporal gradient of cdk1-cyclinB, whereby gradual lifting of inhibition by cdk1-cyclinB leads to a progressive activation of RhoA. RhoA inactivation then results from a negative feedback downstream of RhoA. Thus, the spatiotemporal coupling of two conserved molecular modules, cdk1-cyclinB for the cell cycle and RhoA for contractility, gives rise to a transient band of cortical contractility moving across the oocyte.

Our finding that the cdk1-cyclinB gradient drives the SCW extends the classical cell cycle clock model to the spatial dimension. We show that in cells that are large and undergo rapid cell cycle transitions, diffusion is sufficiently slow for a spatial gradient of cell cycle regulators to form. Thus, cell cycle states will not only vary over time, but also spatially. This spatial inhomogeneity of the cell cycle, also called metachronicity, has previously been shown in *Xenopus* oocytes, and has been linked to SCWs, fully consistent with our findings[17, 20]. However the origin of the spatial inhomogeneity remained unclear in the non-transparent *Xenopus* oocytes. Here, we show and directly visualize that the spatial inhomogeneity of the cell cycle is shaped by nuclear accumulation and diffusion of cdk1-cyclinB. Our observation that the speed of the SCW depends on the shape of the confining compartment suggests that cdk1-cyclinB inactivation does not spread across the oocyte as a trigger wave, different from its activation[18].

In somatic cytokinesis, in smaller cells, the same conserved RhoA/Rok/NMYII module is controlled temporally by cdk1-cyclinB, while it is spatially patterned by spindle microtubules[27, 36]. The difference for SCWs is that in a large cell cdk1-cyclinB controls the RhoA module both in space and time, while the spindle is too small to control cell-scale behavior. We speculate that RhoA activation guided by the gradient of cdk1-cyclinB, may be an ancestral module in cell division, coarsely positioning the RhoA activity zone in large cells, while in somatic cells this process is further refined by mechanisms that target RhoA specifically to the spindle midzone[37, 38]. Our work also revealed a novel negative feedback downstream of RhoA restricting the contraction to a transient band. In the future, it will be interesting to explore the molecular details of this negative feedback possibly acting through Rok and GAPs[39, 40], or potentially involving a mechanosensitive component directly reacting to the tension generated by the contraction, potentially through mechanosensitive GAPs, such as p190RhoGAP[41]. It will also be interesting to see how the global RhoA SCW wave shaped by cdk1-cyclinB is combined with the embedded and smaller-scale dynamics of the RhoA module showing excitable behavior[26, 40], which may further contribute to patterning the RhoA activity zone.

The two highly conserved molecular modules, cdk1-cyclinB for cell cycle control and RhoA/Rok/NMYII for contractility, function in all dividing animal cells. Therefore, all cells carry the potential for the behavior we describe here. Our model predicts that whether and how these cortical behaviors manifest themselves depend largely on cell size and geometry. This offers an elegant explanation for the diversity in appearance and timing of SCWs in oocytes and embryonic cells of different species. Diversity likely arises from the different cdk1-cyclinB gradient shapes dependent on cell and nuclear size and geometry as well as cell cycle timing—similar to our observations in oocytes with artificially changed shapes. It is likely that this diversity of SCWs has been harnessed for a wide range of functions in oocytes and embryos, such as localizing developmental determinants[4–6, 8, 9].

## Methods

**Oocyte collection and treatments**. We used oocytes of bat stars (*Patiria miniata*) obtained from Southern California Sea Urchin Co., Marinus Scientific, South Coast Bio-Marine, or Monterey Abalone Co. The animals were kept in seawater tanks at 15 °C at the European Molecular Biology Laboratory (EMBL) Marine Facility. Oocytes were isolated from ovaries, and injection of oocytes was performed in custom-made chambers using mercury-filled needles[42, 43]. Oocytes injected with mRNAs were incubated overnight at 14 °C, or for 48 h to achieve overexpression; imaging was performed at 20 °C. Oocyte maturation was induced by treatment with 10 µM 1-methyl adenine (ACROS Organics). All presented data sets are gathered from oocytes from multiple collections.

**Drug treatments**. Inhibitors dissolved in DMSO were diluted in sea water to the indicated final concentration, and oocytes were incubated in this solution for the times given below before hormone addition. These inhibitors were: Cytochalasin D (Sigma; 10 mM, incubated for 40 min), Nocodazole (Sigma; 3.3 µM, incubated for 5 min), ML-7 (Enzo; 100 µM, incubated for 40 min), and Blebbistatin (Abcam; 300 µM, incubated for 1 h). Inhibitors dissolved in water were injected into the oocytes as above approximately 10 min after NEBD. These inhibitors were: Phalloidin (Invitrogen; 230 pg), Y-27632 (Enzo; 12 mM), and C3 exoenzyme (Enzo; 150 pg). For local cdk1 inhibition, 2 µl of the concentrated stock of Roscovitine (Merck Millipore; 50 mM) or RO-3306 (Santa Cruz, 10 mM) were applied directly to the oocytes in chambers with one open side 25–30 min after NEBD. Active cdk1-cyclinB complex isolated from starfish oocytes was a kind gift of Eiichi Okumura and Takeo Kishimoto[44], and approx. 200 pg of the protein was injected in a 1:1 mix with 20 mg/ml 500 kDa dextran-Cy5 (Invitrogen) 10 min after NEBD.

---

**Fig. 3** cdk1-cyclinB forms a spatiotemporal gradient controlling the RhoA module. **a** Oocyte expressing RhoA-GTP marker EGFP-rGBD and locally treated with DMSO (as in scheme to the left), starting point of SCW marked by a red asterisk. Right: Kymograph of the cortical rGBD signal. Scale bar=20 µm. **b** Same as **a** except oocyte was globally treated with the cdk1 inhibitor RO-3306. **c** Same as **a** except oocyte was locally treated with the cdk1 inhibitor RO-3306, the starting point of the SCW is marked by a red asterisk. **d** Pseudo-colored frames from a time-lapse recording of an oocyte expressing cyclinB-EGFP during meiosis I. Right: contrast adjusted to visualize decreased levels of cyclinB during SCW. Images are 15 s averages of 1.5 s per frame recording. Scale bar=20 µm; time relative to NEBD, in mm:ss. See also Supplementary Movie 5. **e** Quantification of cyclinB-mEGFP intensities of the oocyte shown in **d**. Left: kymograph of the subcortical cyclinB-EGFP fluorescence intensity during meiosis with intensity-isolines in white. Middle: Zoom of the white dashed box including the time of the SCW with adjusted contrast and the isoline conforming to the SCW highlighted in red. Right: kymograph of radii of curvature during the same time window with the isoline from the middle plot overlaid. **f** Kymograph of the simulated cdk1-cyclinB concentration profile along the cortex. For details of the simulation see Methods. **g** The reaction system of cdk1-cyclinB inactivation. **h** Frame of the 3D finite element simulation of the cdk1-cyclinB reaction-diffusion system. **i** For the oocyte shown in **d**, cyclinB-EGFP subcortical intensities plotted over time at angles from animal pole to vegetal pole (darkest to lightest gray), smoothed data shown in bold, raw data in thin line. Red line same as isoline in **e**. **j** For the oocyte shown in **d**, subcortical cyclinB-EGFP intensities plotted from the animal to vegetal pole during the SCW at time points 40, 43, 46, 49, 52, and 55 min after NEBD (darkest to lightest gray). Red line same as isoline in **e**. **k** Simulated cdk1-cyclinB intensity as in **i**. **l** Simulated cdk1-cyclinB intensity profiles as in **j**

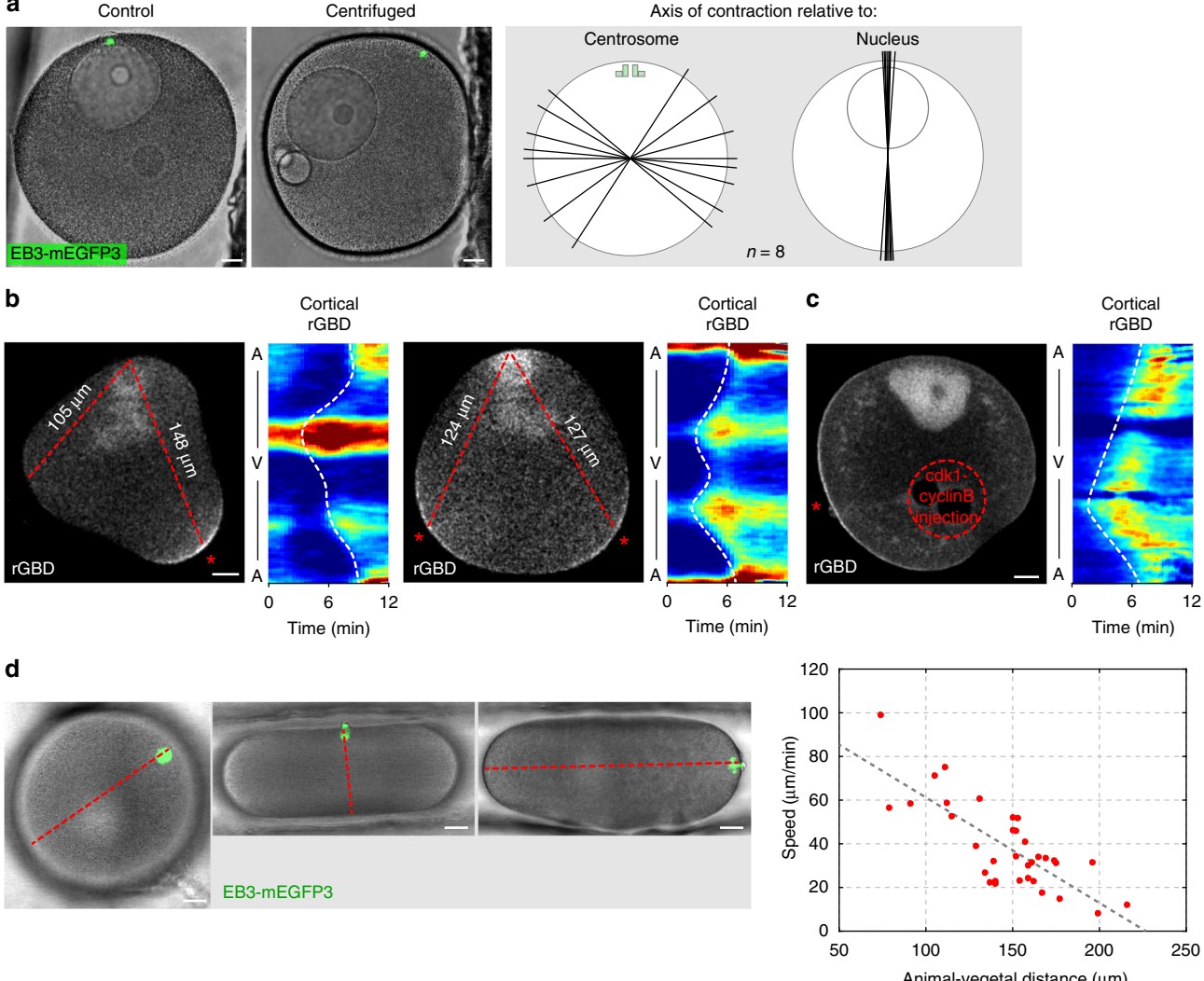

**Fig. 4** The SCW front is guided by the cdk1-cyclinB gradient. **a** Left panel: control oocyte (left) and oocyte after centrifugation (right), with centrosomes position indicated by microtubule label EB3-3mEGFP (green) marking the original position of the animal pole. Scale bars=20 μm. Right panel: schemes showing the axis of SCWs in individual centrifuged oocytes (one line per oocyte) relative to the position of centrosomes and nucleus, respectively. **b** Selected frames from time-lapse recordings of two oocytes shaped into triangles by microfabricated chambers, and expressing RhoA-GTP marker rGBD-EGFP. Red dashed lines indicate the distance between the animal pole (top) and the furthest corner(s) of the oocyte. The starting points of the SCW are marked with a red asterisk. The respective kymographs show cortical rGBD-EGFP fluorescence intensities during the SCW. Scale bars=20 μm. **c** Selected frame from a time-lapse recording of an oocyte expressing rGBD-EGFP and injected with active cdk1-cyclinB protein to the area indicated by the red dashed circle. Red asterisk indicates the starting point of the SCW. Respective kymograph shows cortical fluorescence intensities for rGBD-EGFP during the SCW. Scale bar=20 μm. **d** Oocytes placed in microfabricated chambers leaving shape unchanged, compressed or expanded along the animal-vegetal axis, respectively. The animal pole is marked by the spindle visualized by EB3-3mCherry. Red dashed line indicates the animal-vegetal axis. Scale bar=20 μm. Quantification of the effects of these shape changes on SCW speed plotted as the speed of the SCW against animal-vegetal distance with each dot representing an individual oocyte

**Physical manipulations and surface tension measurements**. Oocytes were centrifuged by placing the cover slip holding the oocytes in custom made holders in 50 ml Falcon tubes and centrifuged at 420 g for 45 min at 4 °C[45].

Shape changes were done by placing oocytes in metaphase, 30 min after hormone addition, into PDMS chambers of different shapes. PDMS microfabricated chambers were produced as described in ref. [35], adapted to the size of the starfish oocyte.

The jelly coat was removed either by enzymatic treatment with Actinase E (Sigma-Aldrich) at 0.1 mg/ml for 1 h[23], or by treatment with sea water pH 4 for 5 min[22]. Surface tension measurements were performed using suction pipettes with a diameter of 60 μm on oocytes in metaphase[24]. The surface tension was calculated using the Young–Laplace formula:

$$T_c = \frac{\Delta P}{2\left(\frac{1}{R_P} - \frac{1}{R_C}\right)}$$

where $T_c$ is the surface tension of the cortex, $\Delta P$ is the pressure difference across the interface, $R_P$ is the radius of the pipette and $R_C$ is the radius of the cell.

**Fluorescent constructs and confocal microscopy**. To create fluorescent constructs, the sequence of the protein of interest was identified in a translated transcriptome dataset (http://www.lenartlab.embl.de:4567/), and cloned from a cDNA library or synthesized by GENEWIZ[43]. The fluorescent constructs used were all cloned into pGEMHE vectors tagged with mEGFP or mCherry[43]. Synthetic mRNAs were produced following linearization using the AmpliCap-Max™ T7 High Yield Message Maker Kit (CellScript) and A-Plus™ Poly(A) Polymerase Tailing Kit (CellScript). The constructs used were mEGFP-non-muscle myosin II heavy chain, myosin regulatory light chain-mEGFP, and cyclinB-mEGFP. Other reporter constructs used were Rhotekin GTP-binding domain (EGFP-rGBD, *Mus musculus*)[25], Utrophin-CH domain-mEGFP[46], gifts from William M. Bement, hsEB3-mEGFP[47] and histone-2B-EGFP (H2B-EGFP)[48].

Imaging was performed on a Leica SP5 confocal microscope using a 1.1 NA HC PL APO ×40 water immersion objective, and equipped with a fast Z-focusing device (SuperZ Galvo stage) (Leica Microsystems). Imaging was commonly performed in multi-position mode at a speed of 600 Hz, with a time resolution of 10 s, zoom of 1.3 and 2-times line averaging, using HyD hybrid detectors. Only oocytes oriented with their AP-VP axis perpendicular to the light path were imaged.

**Image analysis and quantifications**. Image analysis was done using ImageJ and Matlab (Mathworks) using the Miji extension. To determine the curvature change during the SCW, frames of time lapse recordings were first segmented using a sparse-field level set algorithm by minimizing the Chan–Vese energy function[49, 50]. The segmented outline was smoothed by fitting a piecewise polynomial to the outline in polar coordinates and the first principal curvature was then calculated for small segments (each around 2 μm) of the segmented outline using finite differences (Supplementary Fig. 1A). To remove uneven starting shapes of oocytes, relative curvatures were calculated by subtracting the curvature of the first image. To quantify the strength of the SCW, the variance of the radii of curvature values during the SCW was calculated and the background curvature variance was subtracted (measured during an equal time window in metaphase). The cortical and subcortical fluorescence intensity was measured by creating a ring based on the segmented outline and measuring the fluorescence intensity in the same (sub)cortex segments used for calculating the curvature. The kymographs for both the simulation of the cyclinB gradient and the SCW were generated using the same algorithm as described above on the individually simulated frames. The polar body area was measured manually at its point of furthest protrusion in ImageJ.

Plotting of data was done using R and Matlab, using boxplots where middle line is the median, box includes data between 25th and 75th percentile and whiskers show highest and lowest data point excluding outliers. Each treatment condition was tested and quantified on between 5 and 15 oocytes from multiple collections and controls were performed on the same day on oocytes from same collection. No randomization or blinding was performed. Statistical significance was calculated using ANOVA (Microsoft Excel Analysis ToolPak).

**Computer simulations**. For 3D simulations the oocyte's surface was approximated by a triangular mesh that is locally contracted by a surface tension band shaped as a half sine period with the width determined by the points at which the function goes through zero. The tension is balanced by in-plane elastic forces, out-of-plane bending forces, and osmotic forces in the volume. The complete Hamiltonian reads:

$$H_{shape} = \oint \left(2\kappa_b H^2 + \sigma(l)\right)\mathrm{d}A + \oint \left(\frac{K_\alpha}{2}\alpha^2 + \mu\beta\right)\mathrm{d}A_0 + \frac{k_V}{2}(V - V_0)^2,$$

where $\kappa_b$ denotes the bending stiffness of the membrane, $H$ the mean curvature, $\sigma$ the surface tension symmetrical around the A–V-axis, $l$ the arc length along the contour of the axisymmetric shape, which is measured on the curve resulting from intersecting the surface with a plane containing the axis of symmetry. $K_\alpha$ and $\mu$ are the elastic stretch and shear moduli, $\alpha$ and $\beta$ the elastic invariants associated with stretch and shear (see ref. [51]), $\mathrm{d}A_0$ a surface element of the undeformed surface, $k_V$ the volume stiffness due to osmotic forces, and $V$ and $V_0$ the actual and the uncontracted volume. The parameter used for the width of the contracting band was determined so that the observed resulting deformation closely matched that observed during the SCW, and set at 100 μm. For further parameter values see Supplementary Table 1. The effect of the surface contraction is two-fold. First, a contractile ring generates a force that points at its center. Second, a tension gradient leads to a local imbalance of forces on neighboring sites at the membrane, dragging material up the gradient. These effects are balanced by long-ranged elastic and osmotic effects, which result in a global shape change in response to a more localized position of the surface contraction band. The shape problem was solved for each time point independently at steady state. Kymographs of simulated curvature radii were constructed from the simulation results at the different time points using the same image processing pipeline as for experimental kymographs.

The cdk1-cyclinB gradient was simulated in the finite element software COMSOL (COMSOL Inc.) in order to identify the processes relevant for the establishment of the cdk1-cyclinB gradient. The model starts at NEBD, when the initial conditions state a high concentration of cdk1-cyclinB in the nucleus. As formalized in the equations below, cdk1-cyclinB diffuses, is globally degraded and locally produced in the nuclear region. Further, cdk1-cyclinB interacts with APC/C, whose activation is cdk1-cyclinB dependent:

$$\dot{cdk1} = D_{cdk1} \cdot \Delta cdk1 - \left(k_0 + k_1 \frac{APC}{J^2 + cdk1^2}\right) \cdot cdk1 + k_2\,\Theta\left(R_{nucleus}^2 - (\vec{x} - \vec{r}_{nucleus})^2\right)$$

$$\dot{APC} = D_{APC} \cdot \Delta APC - k_3 APC + k_4 \frac{APC^2}{K^2 + APC^2} cdk1$$

For parameter values used see Supplementary Table 2, $\Theta$ here is the Heaviside step function.

**Data availability**. The data sets generated in the course of this study are available upon request from the corresponding author. Similarly, the Matlab algorithm for segmentation, curvature and cortical intensity measurements, as well as the code for simulation of SCW and cdk1-cyclinB profile, is available upon request from the corresponding author. The starfish oocyte transcriptome dataset used are available at: http://www.lenartlab.embl.de:4567/.

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

## Acknowledgements

We would like to thank Jan Ellenberg (EMBL) and Nils Klughammer (BioQuant) for comments on the manuscript. We would like to acknowledge Nicolas Minc (Institut Jacques Monod) for help with microfabrication, Kinneret Keren (Technion) for advice, William M Bement (University of Wisconsin-Madison) for advice and plasmid constructs, Eiichi Okumura and Takeo Kishimoto (Tokyo Institute of Technology) for purified active cdk1-cyclinB protein, Ulla-Maj Fiuza (EMBL) and Jean-Léon Maître (Institute Curie) for help with pipette suction measurements. We would like to additionally acknowledge the essential support of EMBL's Advanced Light Microscopy Facility and Laboratory Animal Resources, Kresimir Crnokic in particular. The work was supported by the European Molecular Biology Laboratory, and specifically by the EMBL International PhD Programme to J.B. U.S.S. is member of the cluster of excellence CellNetworks and of the Interdisciplinary Center for Scientific Computing (IWR) at Heidelberg.

## Author contributions

J.B. and P.L. designed experiments, J.B. performed experiments and analyzed the data, C.A.B. and U.S.S. designed models and C.A.B. performed the simulations, I.M. produced image analysis script, M.M., K.S. and S.T. generated constructs. J.B., P.L., C.A.B. and U.S.S. wrote the manuscript.

## Additional information

**Competing interests:** The authors declare no competing financial interests.

**Change history:** An incorrect version of the Supplementary Information was inadvertently published with this Article where the wrong file was included. The HTML has been updated to include the correct version of the Supplementary Information.

