## [Peer Review File · Nature Communications]

Reviewers' Comments:

Reviewer #1:

Remarks to the Author:

Review of "Surface contraction waves are driven by a traveling band of RhoA-activated myosin II guided by a cdk1 gradient" by Bischof et al.

Surface contraction waves (SCWs) are cortical contractions that are somehow entrained to cell cycle progression. They were identified in oocytes, eggs and zygotes of a variety of species by several labs in the 1970s and 1980s, and they attracted much attention as a "read out" for m-phase (mitosis or meiosis). That is, SCWs could reliably predict the onset of M-phase in the absence of any other kind of marker. This observation spawned a series of experiments in which the SCWs were exploited to reveal the basic mechanisms of cell cycle regulation in eggs and embryos. Following the discovery of the conserved roles of cdk1 and cyclin in mitotic entry, the field moved on to characterize the upstream mechanisms of cdk1 activation without ever asking or answering a fundamental question: what, exactly, are surface contraction waves? That is, how is it that cdk1, whose rise is associated with and required for SCWs, coupled to changes in the cell cortex? This importance of this question goes well beyond the SCWs themselves however, in that the precise mechanisms by which cell cycle changes are linked to changes in the the cell cortex are, in general, very poorly understood.

In the current study, the authors have addressed the fundamental question alluded to above by investigating SCWs in starfish oocytes. Using a combination of high resolution imaging, molecular, mechanical and pharmacological approaches, they have revealed that SCWs arise from a wave-like increase in the activity of Rho, a small GTPase. This wave traverses the cortex from the vegetal pole to the animal pole, and moves in its wave like manner as a combination of cdk1 inactivation (which relieves Rho inhibition) at its leading edge and negative feedback from rock (rho dependent kinase) at its trailing edge. The contraction itself is a consequence of myosin-2 activation by Rho.

This is a truly beautiful piece of work both in the traditional sense of the word (the images and movies are gorgeous) and in the sense that the experiments were well-designed and executed, the work is thorough, and a 40-year-old mystery has been solved. Part of this stems from the attention paid by the authors to biophysical quantification—to the best of my knowledge, all previous studies of SCW treated them as simple yes-or-no phenomena rather than complex mechanical processes. This has allowed the authors to demonstrate via both imaging and modeling that the SCW results not from a simple wave front of contraction (the simplest model) traveling from the vegetal to the animal pole, but a wave-like band of contraction. It has also allowed the authors to directly map the band of increased Rho activity simultaneously with the SCW itself, and to subsequently show that suppression of rock activity converts the band into a simple wave front, providing a simple explanation for the negative feedback.

The cdk1 part of the story also benefits from the care and skill of the authors. The authors were able to overlay their physical SCW map with a map of Cdk1 inactivation (as judged by loss of cyclin signal). In addition, the authors conduct clever experiments involving manipulation of the cdk1 gradient by various means. These are the proverbial icing on the cake as they both confirmed their basic model and explained one of the more curious features of SCWs—their invariant movement from the vegetal to the animal pole of the cell.

I have only one or two very minor comments to make:

1. Given the conservation and importance of their players, it seems likely that the SCWs are not confined to oocytes, eggs and zygotes but are just easiest to detect in such models. The authors may wish to make this point clear, lest it be assumed that the behaviors they describe are restricted to oocytes and zygotes.

2. Before referring to the target of the rGBD-based probe as “active RhoA” the authors should check to make sure that RhoA is much more abundant in their cells than RhoB, which is also detected with this probe. If the two are equally abundant, it is probably best to just refer to “active Rho” rather than “active RhoA”.

3. The purist would probably object to EB3 being referred to as a marker for centrosomes since it does not directly label centrosomes (in contrast to something like centrin) but rather the plus ends of rapidly growing microtubules. This sounds picayune in the extreme, but referring to it as a marker of centrosome position is probably more accurate.

Reviewer #2:

Remarks to the Author:

Bischof et al. present an elegant, well-written and concise story that revisits a classical developmental biology phenomenon, surface contraction waves, using modern cell biology tools and mathematical modeling. Their paper provides novel experimental results and valuable insight into the molecular mechanisms of this exciting phenomenon. The combination of excellent imaging, thoughtful and innovative experimental perturbations together with modeling marks this contribution as a significant novel advance in the field and provides for an interesting reading with captivating images and movies.

1. The authors convincingly show that Cdk1-CycB activity is inhibitory to the SCW and allude to the published results that CDK phosphorylation is inhibitory to the Rho GEF Ect2. Do the authors have any direct evidence that Ect2 is the GEF responsible for the activation of RhoA in the SCW? If so, these results should be presented. If, for whatever reason, demonstration of direct involvement failed, this should be mentioned explicitly.

2. The authors present tantalizing results that show that inhibition of Rho kinase (which one? I or II, or the cells in question possess only one Rok?) transforms an apparently excitable RhoA wave into a bistable switch wave (I am guessing from the phenotype). What is their hypothesis of the molecular mechanism of RhoA inhibition by Rok?

3. Modeling of the mechanical wave. It is not clear how the authors are modeling propagation of the wave of contraction in their model. Methods only provide a Hamiltonian function. How was it used to perform dynamic simulations? If dynamic simulations were not performed, how was the kymograph that is shown in Figure 1c produced? How was the size of the contraction wave determined in the model?

4. Somewhat disappointingly, the Discussion ends very abruptly on a rather weak sentence: “This global pattern of contractility by... is distinct from the concept of trigger wave in excitable media...” How? In what way? The paper is replete with interesting results whose novelty is unquestionable. It is not clear why would the authors want to end on a strange confrontational note. First of all, it is clear that the chemo-mechanical wave the authors describe IS excitable in nature. Second, the authors put an effort to demonstrate that the wave is a result of spatial dependence of cellular parameters, i.e., the activity of CDK that in the past has been largely considered spatially-homogeneous and only time-dependent. However, in this last sentence they claim that their observations are determined by global rather than local properties. This obvious contradiction raises eyebrows and questions the intentions of this last sentence. It is clear that SCW is distinct from the waves described by, e.g., Bement et al, 2015, but the sentence is poorly constructed and attempts to cram too much in too little a space. If the authors must contrast their findings to those of others, of which I see no need as the differences are quite clear, they should do it properly and clearly argument the differences in several well-developed sentences. As it is now, the last sentence only diminishes the excellent impression produced by the rest of the paper. This is very unfortunate.

Style and language issues:

1. On page 3, Supplementary figure 1 is cited before the actual Figure 1. This is somewhat awkward and with minor alterations could be avoided. On this same page, there is much juggling between Figure 1 and Supplementary Figure 1. Perhaps, some of the material can be moved from

the supplement into the actual figure 1 to avoid this?

2. Discussion. First sentence reads: "In conclusion, ... the classical cell cycle clock model of the cell cycle..." This wording is awkward. Please reformulate to avoid repetition of "cell cycle" in close proximity.

3. Generally in the literature, there is an atrocious abuse of expression "taken together". This article uses this parasite sentence also inordinately frequently. It is okay to say "taken together, our results demonstrate..." but it is not possible to say "taken together, we demonstrate" (two instances: last paragraph of Discussion and last sentence of Introduction). I am sure the authors don't mean that they should be taken together? These two sentences look absolutely fine if "Taken together" is removed from their beginning. Please correct.

Reviewer #3:

Remarks to the Author:

Surface contraction waves (SCWs) are conserved stereotypical changes in oocytes and embryos that lead to large-scale shape changes. They are coupled to cell cycle transitions and spatially coordinated with the cell axis. In this manuscript, Bischof and collaborators nicely and convincingly describe this cortical contraction band in metaphase I in starfish oocyte and decipher the molecular mechanism at play originating and regulating it (in space and time). Overall it is a very comprehensive study. Another strength of this paper is the use of state of the art techniques, allowing the authors to tackle their biological question from different and complementary angles, combining quantitative imaging, biochemical and mechanical perturbations with mathematical modelling. The paper is also very well written with a large amount of high-quality data presented in the figures.

In conclusion, this is a high quality research. The findings are novel and very interesting. I have the following minor concerns/comments:

1- What is the role of this SCW in metaphase I in starfish? Is it essential for polar body extrusion? In the conditions where the SCW is altered (in space and/or time), does it impede polar body extrusion? And on chromosome segregation?

2- Why removing the jelly softens the oocyte? Is the jelly tightly connected to the oocyte cortex (via physical contacts), acting like a scaffold? Does it change the cortex of the oocyte itself (composition)? Does it change the shape of the cell (in Fig 1e oocytes without jelly appear very deformed compared to the ones in Fig S1 d)? Also, could the authors do the reverse and increase cortical tension to test if it decreases the strength of shape change (stiffen it using Concanavallin A as in Kunda P Curr Biol 2008, or embed their oocytes in collagen) and compare to the prediction of their model?

3- The depletion of NMYII under the cortex in Fig 2b is not very convincing. Could the authors measure fluorescence intensity instead of showing kymographs? And why would its recruitment at the cortex deplete the subcortical pool, the overexpressed probe should be in excess no?

4- In Fig 2f, why ML-7 has no effect? Do the authors have a read-out of the activity of NMYII after ML-7 addition (in other words are they sure that their ML-7 treatment is efficient in inhibiting myosin light chain kinase)?

5- The authors developed a reaction-diffusion model that incorporates cyclinB degradation by APC/C and nuclear concentration of Cdk1-cyclinB. The assumption is that Cdk1-cyclinB is only regulated by the rate of cyclinB degradation. However, post-translational modifications (namely phosphorylations/dephosphorylations) are known to be important in regulating Cdk1-cyclinB activity. Could these regulations play a role in the local activation or inhibition of Cdk1-cyclinB activity to initiate the SCW, and could the authors implement in their reaction-diffusion model such

events?

6- On the same line, to further prove that the degradation of cyclinB is key to generate the cdk1-cyclinB gradient, could the authors express non-degradable cyclinB in their oocytes (to test if it suppressed the gradient and SCW) and/or modulate SCW speed by overexpressing WT cyclinB at different concentrations (as in Fig S3a, but with different concentrations, a way to modulate the gradient in time and slow down SCW in a dose dependent manner)?

Point-by-point response to the Reviewers

Reviewer #1:

Surface contraction waves (SCWs) are cortical contractions that are somehow entrained to cell cycle progression. They were identified in oocytes, eggs and zygotes of a variety of species by several labs in the 1970s and 1980s, and they attracted much attention as a “read out” for m-phase (mitosis or meiosis). That is, SCWs could reliably predict the onset of M-phase in the absence of any other kind of marker. This observation spawned a series of experiments in which the SCWs were exploited to reveal the basic mechanisms of cell cycle regulation in eggs and embryos. Following the discovery of the conserved roles of cdk1 and cyclin in mitotic entry, the field moved on to characterize the upstream mechanisms of cdk1 activation without ever asking or answering a fundamental question: what, exactly, are surface contraction waves? That is, how is it that cdk1, whose rise is associated with and required for SCWs, coupled to changes in the cell cortex? This importance of this question goes well beyond the SCWs themselves however, in that the precise mechanisms by which cell cycle changes are linked to changes in the cell cortex are, in general, very poorly understood.

In the current study, the authors have addressed the fundamental question alluded to above by investigating SCWs in starfish oocytes. Using a combination of high resolution imaging, molecular, mechanical and pharmacological approaches, they have revealed that SCWs arise from a wave-like increase in the activity of Rho, a small GTPase. This wave traverses the cortex from the vegetal pole to the animal pole, and moves in its wave like manner as a combination of cdk1 inactivation (which relieves Rho inhibition) at its leading edge and negative feedback from rock (rho dependent kinase) at its trailing edge. The contraction itself is a consequence of myosin-2 activation by Rho.

This is a truly beautiful piece of work both in the traditional sense of the word (the images and movies are gorgeous) and in the sense that the experiments were well-designed and executed, the work is thorough, and a 40-year-old mystery has been solved. Part of this stems from the attention paid by the authors to biophysical quantification—to the best of my knowledge, all previous studies of SCW treated them as simple yes-or-no phenomena rather than complex mechanical processes. This has allowed the authors to demonstrate via both imaging and modeling that the SCW results not from a simple wave front of contraction (the simplest model) traveling from the vegetal to the animal pole, but a wave-like band of contraction. It has also allowed the authors to directly map the band of increased Rho activity simultaneously with the SCW itself, and to subsequently show that suppression of rock activity converts the band into a simple wave front, providing a simple explanation for the negative feedback.

The cdk1 part of the story also benefits from the care and skill of the authors. The authors were able to overlay their physical SCW map with a map of Cdk1 inactivation (as judged by loss of cyclin signal). In addition, the authors conduct clever experiments involving manipulation of the cdk1 gradient by various means. These are the proverbial icing on the cake as they both confirmed their basic model and explained one of the more curious features of SCWs—their invariant movement from the vegetal to the animal pole of the cell.

We thank the Reviewer for their very positive evaluation of our work, and we are very glad to see the shared excitement about the mechanisms underlying this striking and widely conserved cellular behavior.

I have only one or two very minor comments to make:

- 1. Given the conservation and importance of their players, it seems likely that the SCWs are not confined to oocytes, eggs and zygotes but are just easiest to detect in such models. The authors may wish to make this point clear, lest it be assumed that the behaviors they describe are restricted to oocytes and zygotes.*

We thank the Reviewer for raising this important point. We amended the Discussion to include these questions explicitly (see text from line 231).

Indeed, both molecular modules involved, cdk1-cyclinB for cell cycle control and RhoA for contractility, are highly conserved and functional in all dividing animal cells. Thus, the mechanisms described here could in principle be present in all cells. However, the larger the cell and the more asymmetric the cdk1-cyclinB source, the more likely it should be that SCWs occur. In this sense, oocytes with their large size and asymmetrically located nucleus are ideal. The cdk1-cyclinB gradient is further influenced by the diffusion constant, which in starfish oocytes is expected to be low due to yolk platelets crowding the cytoplasm (Terasaki et al. 2003; Terasaki et al. 2001). Consistent with this view, we observed weaker and more subtle contraction waves during the early cleavage divisions in the starfish embryo with a centrally located nucleus.

Based on the above, we expect to see SCW in all oocytes with strength and prominence dependent on the size and organization of the particular oocyte, its nucleus and cytoplasm. This means that while SCWs have so far been described in large oocytes, they may have been overlooked in smaller ones (e.g. mouse). Additionally, it is interesting to speculate if a cdk1-cyclinB gradient may build up in somatic cells. However, our model predicts that such effects, if present at all, are likely to be very subtle, as somatic cells are much smaller and typically round up during division, with the nucleus located in the center. Furthermore, diffusion constants in somatic cells are higher than in the starfish oocyte. This in combination of the smaller cell size will further decrease the slope of the gradient.

2. Before referring to the target of the rGBD-based probe as “active RhoA” the authors should check to make sure that RhoA is much more abundant in their cells than RhoB, which is also detected with this probe. If the two are equally abundant, it is probably best to just refer to “active Rho” rather than “active RhoA”.

Searches of starfish sequence databases, specifically of oocyte transcriptome datasets, reveal that starfish only possess a single Rho mRNA and thus protein, which we refer to as RhoA for consistency with the literature. We now amended the text to indicate this more clearly (see text line 97). We now also refer to our own sequence database which is publically available in the Methods section.

Starfishes are echinoderms, which belong to the deuterostome lineage of metazoan phylogeny and have a molecular complement similar to vertebrates, as illustrated for example in our recent work on centrioles (Borrego-Pinto et al. 2016). Importantly, however, echinoderms did not undergo genome duplication(s) like the vertebrate branch of deuterostomes. Therefore, it is typical to echinoderms that only one gene is present, where vertebrates have two or three. This is true for all of the proteins investigated in this study, i.e. starfish only have a single Rho, Rho-kinase as well as a single NMYII heavy chain. Another very illustrative example is the single starfish Aurora kinase that performs all functions carried out by Aurora A/B/C in vertebrates (Abe et al. 2010).

3. The purist would probably object to EB3 being referred to as a marker for centrosomes since it does not directly label centrosomes (in contrast to something like centrin) but rather the plus ends of rapidly growing microtubules. This sounds picayune in the extreme, but referring to it as a marker of centrosome position is probably more accurate.

We thank the Reviewer for pointing out this issue. We agree that the terminology chosen was imprecise and have amended the legend of Figure 4a accordingly.

Reviewer #2:

Bischof et al. present an elegant, well-written and concise story that revisits a classical developmental biology phenomenon, surface contraction waves, using modern cell biology tools and mathematical modeling. Their paper provides novel experimental results and valuable insight into the molecular

mechanisms of this exciting phenomenon. The combination of excellent imaging, thoughtful and innovative experimental perturbations together with modeling marks this contribution as a significant novel advance in the field and provides for an interesting reading with captivating images and movies.

We thank the Reviewer for their very favorable evaluation.

1. The authors convincingly show that Cdk1-CycB activity is inhibitory to the SCW and allude to the published results that CDK phosphorylation is inhibitory to the Rho GEF Ect2. Do the authors have any direct evidence that Ect2 is the GEF responsible for the activation of RhoA in the SCW? If so, these results should be presented. If, for whatever reason, demonstration of direct involvement failed, this should be mentioned explicitly.

The Reviewer raises here a very important question that we extensively tried to address. While we note that our proposed model stands independent of the specific molecular identity of the Rho GEF, we very much would like to identify and characterize the relevant exchange factor. However, as detailed below, fully clarifying this issue will require the development of new methodologies to effectively deplete endogenous proteins, which is beyond the scope of the present study. We have amended the manuscript to include new data that together makes it very likely that Ect2 is the relevant GEF, and explicitly mention the remaining limitations in the text (line 128 onward).

An analysis of the starfish transcriptome databases showed that Ect2 is present in the oocyte, as well as few other potential candidate GEFs, such as GEF-H1 and MyoGEF. However, based on the literature, Ect2 is by far the most likely candidate.

Bill Bement and George von Dassow kindly provided us with an EGFP-labelled version of Ect2, and we were able to confirm all their recently published observations (Bement et al. 2015). That is, overexpression of Ect2 strongly enhances RhoA activity resulting in dramatic activity waves, which we will refer to here as ‘ripples’. It is critical to distinguish these small scale oscillations of RhoA, the focus of the Bement work, from the large scale SCW, which is the topic of our present study: the RhoA ripples occur within the large SCW wave.

The effect of Ect2 overexpression on RhoA ripples suggests Ect2’s involvement, however for the global SCW wave we were only able to detect a small overall increase in the strength of the shape change that remained statistically non-significant (new Supplementary Fig 3b). The likely explanations for this are: first, as we show, the wave front is defined by the cdk1-cyclinB threshold upstream of Ect2, and therefore the timing of RhoA activation, and thus the onset of SCW wave front may not be strongly affected by Ect2 overexpression. Second, there appears to be a limiting component downstream in the Rok-NMYII branch of the signaling pathway, likely NMYII itself, and therefore the strength of the contraction is not strongly increased by over-activation of RhoA following Ect2 overexpression. By contrast, as shown by Bement and coworkers, there appears to be no such limiting component in the other branch of the pathway from RhoA to actin polymerization. Overall, this is consistent with the fact that Bement et al. found no significant effect of NMYII on the RhoA-actin ripples (Bement et al. 2015).

The above reasons combined with the variability inherent to overexpression experiments are the likely explanation for our inability to see significant differences in the SCW strength. Clarifying this issue will thus require development of methods for effective removal and replacement of endogenous Ect2. Our efforts into this direction failed so far, as our attempts to disrupt Ect2 function using a dominant negative form of Ect2 did not impact the SCW, likely due to insufficient expression levels compared to endogenous Ect2. Protein depletion techniques, such as morpholino or siRNA mediated knock down, even if combined with extended incubation of oocytes for up to one week, have so far not provided a reliable depletion of proteins likely due to the exceptional stability of proteins stored in the oocyte. We are currently working on developing alternative methodologies for acutely targeting endogenous proteins for degradation, which may in the future enable us to resolve this important issue.

2. The authors present tantalizing results that show that inhibition of Rho kinase (which one? I or II, or the cells in question possess only one Rok?) transforms an apparently excitable RhoA wave into a bistable switch wave (I am guessing from the phenotype). What is their hypothesis of the molecular mechanism of RhoA inhibition by Rok?

We currently do not have any data on which we can base a molecular hypothesis for the observed negative feedback. The current manuscript focuses on the overall organization and spatiotemporal coupling of cdk1-cyclinB and RhoA modules, and working out the precise molecular wiring internal to the RhoA module will be an exciting topic for future research beyond the scope of the present manuscript.

However we would like to clarify two points: first, we would like to stress that we are not trying to suggest nor have data pointing towards a direct inhibition of RhoA by Rok itself. Published data in different physiological contexts suggests that complex feedback mechanisms may exist linking Rok to GAPs in unexpected ways (Priya et al. 2015; Robin et al. 2016). Further, it is plausible that negative feedback originates from a component downstream of Rok. This component may be NMYII or the negative feedback could involve a mechanosensitive component directly sensitive to the tension generated by the contraction, potentially involving a mechanosensitive GAPs, such as p190RhoGAP (Mammoto et al. 2009). We now included text in the Discussion to discuss these possible mechanisms (see line 223).

Second, the SCW is a ‘wave on top of waves’, which we refer here as ripples, where the oocyte-scale cdk1-cyclinB gradient controls the global SCW wave (this study), within which there are ripples of RhoA (Bement et al. 2015). Thus, as initially proposed by Bement et al. and fully consistent with our data, the global cdk1-cyclinB levels modulate the excitable behavior of RhoA in the ripples. Importantly, we do not think that the SCW wave itself is an excitable (trigger) wave – it is the ripples of RhoA that are excitable waves (see also point 4 below). However, when it comes to the mechanism underlying the negative feedback, it may indeed be necessary to combine these two scales (i.e. wave and ripples), which is a very exciting direction for future research. We also note that we do not claim bistability. The question of trigger versus phase waves has to be discussed separately from the question of bistability. For excitable behaviour, a large excursion in phase space is sufficient, which might result from transient bistability, but it does not have to be (Goryachev et al. 2016). An explanation of our thoughts on the question of excitability and the SCW is now included in the Discussion (see line 214).

To the sub-question related to Rok: searches of starfish sequence databases, specifically of oocyte transcriptome datasets, reveal that starfish only possess a single Rho kinase protein. As for the related question about RhoA by Reviewer 1, we note that starfish are echinoderms and thus did not undergo genome duplication like the vertebrate branch of deuterostomes. Therefore, it is typical that only one gene is present, where vertebrates have two or three. This is true for all of the proteins investigated in this study, i.e. starfish also only have a single RhoA. We have adapted the text to reflect this fact (see line 101).

3. Modeling of the mechanical wave. It is not clear how the authors are modeling propagation of the wave of contraction in their model. Methods only provide a Hamiltonian function. How was it used to perform dynamic simulations?

We agree with the Reviewer that this important methodological point was not clearly explained in the original text. We adapted the Methods section to clarify this (line 333).

In our model, each time point is simulated independently and to steady state, using the mentioned Hamiltonian, and then the individual time points are combined to produce the movies of the simulation. We did not perform dynamic simulations. This ‘adiabatic’ procedure is common in physics and applies

when all subprocesses (here polymerization of actin and recruitment of myosin minifilaments to the cortex) are sufficiently fast that they are in local equilibrium on the time scale of the higher-level process of interest (here SCW). Importantly, although we did not combine the reaction-diffusion model for RhoA and the surface Hamiltonian for the mechanics, we can infer from computer simulations the extent to which the RhoA-activation has to spread in order to generate the observed deformations, thus making sure that our explanations are consistent. In the future, we plan to combine a RD-model along the lines of an earlier model for actomyosin contractility controlled by Rho (Besser & Schwarz 2007) with the mechanical model to recapitulate the whole complexity of the behavior including ‘wave of ripples’, i.e. the excitable nature of RhoA activation.

If dynamic simulations were not performed, how was the kymograph that is shown in Figure 1c produced?

The kymographs were generated by applying the same image analysis pipeline used to calculate surface curvatures of microscope recordings to the output images of individually simulated time points. We have now indicated this in the Methods section (line 307).

How was the size of the contraction wave determined in the model?

The size of the contracting band in the simulations was determined by testing possible values for this parameter to reach a shape change similar to those observed in oocytes. As shown in Letter Fig 1, the shape resulting from simulations is rather sensitive to this parameter: narrowing the band quickly results in very sharp curvature changes not observed in oocytes (see example width of 50 μm), on the other hand, as the width of the band is nearing the oocyte diameter (180 μm) the shape change is much weaker than observed *in vivo* (see example width of 150 μm). We thus choose a band width of 100 μm giving results similar to observed shape changes. As a side note, this indicates that the time scale of the negative feedback defining the width of the band is critical for shaping the SCW. We have amended the text of the methods section to reflect this point (see line 326).

4. Somewhat disappointingly, the Discussion ends very abruptly on a rather weak sentence: “This global pattern of contractility by... is distinct from the concept of trigger wave in excitable media...” How? In what way? The paper is replete with interesting results whose novelty is unquestionable. It is not clear why would the authors want to end on a strange confrontational note. First of all, it is clear that the chemo-mechanical wave the authors describe IS excitable in nature. Second, the authors put an effort to demonstrate that the wave is a result of spatial dependence of cellular parameters, i.e., the activity of CDK that in the past has been largely considered spatially-homogeneous and only time-dependent. However, in this last sentence they claim that their observations are determined by global rather than local properties. This obvious contradiction raises eyebrows and questions the intentions of this last sentence. It is clear that SCW is distinct from the waves described by, e.g., Bement et al, 2015, but the sentence is poorly constructed and attempts to cram too much in too little a space. If the authors must

contrast their findings to those of others, of which I see no need as the differences are quite clear, they should do it properly and clearly argument the differences in several well-developed sentences. As it is now, the last sentence only diminishes the excellent impression produced by the rest of the paper. This is very unfortunate.

We thank the Reviewer for their comment. First, the Reviewer is right about stating that we attempted “to cram too much in too little a space”, but very importantly our intention was exactly the opposite of making a “confrontational note”. Instead, we tried to integrate the work by Bement et al. and our study, and point toward the exciting direction of combining these observations made at two spatial scales and at two levels in the same signaling pathway in the same model system. As a response to the Reviewer’s comment we have rewritten and expanded the Discussion, and would like to thank the Reviewer again for pointing out the poor phrasing leading to this unfortunate confusion.

We here want to take the opportunity to clarify and expand on two important points:

First, the work by Bement et al. and our work are clearly distinct, but as we are studying the same system at two levels, it is a very intriguing possibility to integrate these two scales. Indeed it may be necessary to integrate these two levels, the oocyte-scale SCW wave and the RhoA ripples within, if we aim to understand the full complexity of RhoA activation and the negative feedback, which in turn downregulates RhoA. As beautifully demonstrated by Bement et al., RhoA shows an excitable behavior in this system resulting in the RhoA and F-actin ripples, and they also showed that this excitable behavior is tuned by cdk1-cyclinB. In this sense, we are studying an excitable system, but this does not automatically imply that the SCW wave front would be a moving across the oocyte as a trigger wave in an excitable medium. It is still a remaining and interesting question whether this system will become excitable under certain conditions, as all the required mechanisms (positive and negative feedbacks that can lead to large excursions and possibly bistability) are in place.

Second, although this is not the main focus of our study, we conclude that the wave of cdk1-cyclinB *inactivation* does NOT move across the oocyte as a trigger wave. This is evidenced by the fact that the SCW wave front does not propagate with a constant local velocity once triggered at the vegetal pole (as predicted by the Luther formula for trigger waves), as clearly shown by the experiments in which the shape of the oocytes was manipulated. In these experiments the wave front moves with very different velocities dependent on shape demonstrating the global control of wave propagation (Fig. 4d). This observation is in stark contrast with cdk1-cyclinB *activation* that does spread as a trigger wave, as very clearly demonstrated by the excellent work of Chang and Ferrell in *Xenopus* extracts (Chang & Ferrell 2013), and indeed we have circumferential evidence that this is also the case in starfish oocytes (unpublished data). It is not correct to assume that if cdk1-cyclinB *activation* is a trigger wave then *inactivation* must be a trigger wave as well. In fact, there is no evidence for the latter presented by Chang and Ferrell and the extensive modeling work on cdk1-cyclinB activation and inactivation does not provide support for this either (e.g. Ciliberto et al. 2005; Yang & Ferrell 2013). Again, this is not our main focus, but our data actually does indicate that cdk1-cyclinB inactivation is not spreading as a trigger wave, at least in the starfish oocyte. Therefore, as the SCW is guided by the cdk1-cyclinB-gradient, in physics terms the SCW wave front would rather be considered as phase wave.

Style and language issues:

1. On page 3, Supplementary figure 1 is cited before the actual Figure 1. This is somewhat awkward and with minor alterations could be avoided. On this same page, there is much juggling between Figure 1 and Supplementary Figure 1. Perhaps, some of the material can be moved from the supplement into the actual figure 1 to avoid this?

We thank the Reviewer for the comment and fully agree that the original order of figures and text was not ideal and have changed Fig 1 and Supplementary Fig 1 as well as the surrounding text in the first

paragraph of the Results section to make the flow of the text more straightforward and streamlined (see line 56).

2. Discussion. First sentence reads: "In conclusion, ... the classical cell cycle clock model of the cell cycle..." This wording is awkward. Please reformulate to avoid repetition of "cell cycle" in close proximity.

This was indeed an awkward phrasing and we have changed it to "the classical clock model of the cell cycle" (line 207).

3. Generally in the literature, there is an atrocious abuse of expression "taken together". This article uses this parasite sentence also inordinately frequently. It is okay to say "taken together, our results demonstrate..." but it is not possible to say "taken together, we demonstrate" (two instances: last paragraph of Discussion and last sentence of Introduction). I am sure the authors don't mean that they should be taken together? These two sentences look absolutely fine if "Taken together" is removed from their beginning. Please correct.

We thank the Reviewer for pointing out this unfortunate use of language and have removed the phrase from the text.

Reviewer #3:

Surface contraction waves (SCWs) are conserved stereotypical changes in oocytes and embryos that lead to large-scale shape changes. They are coupled to cell cycle transitions and spatially coordinated with the cell axis. In this manuscript, Bischof and collaborators nicely and convincingly describe this cortical contraction band in metaphase I in starfish oocyte and decipher the molecular mechanism at play originating and regulating it (in space and time). Overall it is a very comprehensive study. Another strength of this paper is the use of state of the art techniques, allowing the authors to tackle their biological question from different and complementary angles, combining quantitative imaging, biochemical and mechanical perturbations with mathematical modelling. The paper is also very well written with a large amount of high-quality data presented in the figures. In conclusion, this is a high quality research. The findings are novel and very interesting.

We thank the Reviewer for this very favorable evaluation.

I have the following minor concerns/comments.

1- What is the role of this SCW in metaphase I in starfish? Is it essential for polar body extrusion? In the conditions where the SCW is altered (in space and/or time), does it impede polar body extrusion? And on chromosome segregation?

We would like to thank the Reviewer for raising this question. Specifically to polar body formation, we first note that initially we were intrigued by the earlier proposal of Hamaguchi and coworkers that the SCW is directly involved in generating the pressure to drive polar body extrusion (Hamaguchi & Hiramoto 1978; Hamaguchi et al. 2007). However, our more detailed studies demonstrated that changing the SCW by either strengthening or weakening it has no effect on the size of the polar body protrusion. We now included these data indicating that polar body extrusion is independent of SCW in Supplementary Figure 2a and refer to these data in Results (line 116). Additionally, we observed normal chromosome segregation in oocytes with stronger contraction waves following MRLC overexpression and normal spindle shapes in oocytes with reduced contraction waves via Rok inhibition (Letter Fig. 2). We prefer not to include these data in the main manuscript as they may be better fitting in our follow up study specifically on polar body extrusion.

As a second point, it is clear that even a small increase in the rate of abnormal oocyte divisions will lead to an increased proportion of aneuploid eggs, thereby directly affecting the reproductive fitness of the species. However, testing whether the SCW has a role in supporting the actual cell division is difficult because, as revealed by our study, the molecular components are shared between the SCW and the cytokinetic ring as well as other contractile processes (now shown in Supplementary Figure 2b, c). Therefore, all treatments affecting the SCW will automatically and unavoidably affect polar body cytokinesis (block ‘pinching off’ of the polar body) that renders the developmental outcome of these experiments hard if not impossible to interpret. For this reason, it remains possible that SCWs do have a role in supporting meiotic divisions.

2- Why removing the jelly softens the oocyte? Is the jelly tightly connected to the oocyte cortex (via physical contacts), acting like a scaffold? Does it change the cortex of the oocyte itself (composition)?

As the Reviewer, we are equally intrigued by the effects of the jelly coat removal, and while it is only of peripheral importance to our present study and its conclusions, it is certainly an exciting topic that we plan to continue to explore in future studies. The effect of jelly coat removal softening the oocytes is very clear, and it cannot only be measured by micropipette suction, but it can be easily observed when microinjecting or handling oocytes.

There is published evidence for a connection between the jelly and the oocyte and therefore a “scaffold”-like function. Ultrastructural investigations of the starfish oocyte indicate that the jelly layer is a tightly interlinked fibrous structure composed of mucopolysaccharids with the oocyte’s microvilli covered in glycocalyx penetrating the vitelline membrane and reaching into the jelly layer (Metz 1985; Schroeder et al. 1979; Endo et al. 1987; Rosenberg et al. 1977). In the ovaries, the jelly is sandwiched between the oocyte and the monolayer of follicle cells, which connect to the oocyte via protrusions reaching through the jelly layer. Upon spawning into the sea, the follicle cells are released and the jelly layer is restructured, likely hardened to provide additional mechanical support to the oocyte in the turbulent environment. All this indicates that the jelly likely acts as a scaffold for the oocyte.

Does it change the shape of the cell (in Fig 1e oocytes without jelly appear very deformed compared to the ones in Fig S1 d)?

In their resting state oocytes will adopt a spherical shape with or without the jelly layer. However, without the jelly layer the oocytes are much more deformable. The larger deformation shown in Fig 1f illustrates the larger shape change during the SCW stronger in oocytes without jelly.

Also, could the authors do the reverse and increase cortical tension to test if it decreases the strength of shape change (stiffen it using Concanavallin A as in Kunda P Curr Biol 2008, or embed their oocytes in collagen) and compare to the prediction of their model?

We tried, as the Reviewer suggests, to increase the cortical tension by various means. We were unsuccessful in adapting the approach by Kunda et al. using Concanavalin A. However, we designed microchambers with exactly matching circular shape, in which the oocytes were efficiently constrained by the surrounding PDMS matrix. In these cases we could no longer observe the shape changes of the SCW – and thus their strength could not be measured as our method for measuring this relies on quantifying shape changes. We could however still visualize the dynamics of active RhoA moving across the oocyte and from this derive the speed of the SCW that is ‘invisible’ on the level of a shape change (as on Fig 4d). The speed measured in this way for embedded oocytes did not significantly differ from controls (Letter Fig. 3).

These observations are fully consistent with our model, in that as opposed to softening the oocytes leading to larger shape changes, strongly stiffening the environment leads to smaller (in this case undetectably small) changes in the cell shape, while the underlying RhoA signaling is still intact.

Letter Figure 3: Oocytes constrained in microchambers

3- The depletion of NMYII under the cortex in Fig 2b is not very convincing. Could the authors measure fluorescence intensity instead of showing kymographs? And why would its recruitment at the cortex deplete the subcortical pool, the overexpressed probe should be in excess no?

We agree with the Reviewer and for clarification we included a fluorescence intensity plot in Figure 2b, and also changed the text to clarify our point (see line 86). That is, the visible depletion of the subcortical pool illustrates that a substantial amount of the NMYII is recruited to the contractile cortex directly from the local subcortical cytoplasmic pool as opposed to NMYII moving along the cortex.

4- In Fig 2f, why ML-7 has no effect? Do the authors have a read-out of the activity of NMYII after ML-7 addition (in other words are they sure that their ML-7 treatment is efficient in inhibiting myosin light chain kinase)?

The Reviewer is correct in noting that we have no independent control to confirm the effectiveness of ML-7, and have changed the text to reflect this (line 103). While we are aware of the fact that ML-7 is by comparison (to for example the Rok inhibitor Y-27632) a compound with low affinity and specificity, we still would prefer to include these data as ML-7 has been widely used and accepted in the literature as an inhibitor of MLCK, including in echinoderm species (Moorhouse et al. 2015). Additionally, we tested another MLCK inhibitor, Pep 18 and consistently observed no effect on the SCW (Letter Fig 4).

5- The authors developed a reaction-diffusion model that incorporates cyclinB degradation by APC/C and nuclear concentration of Cdk1-cyclinB. The assumption is that Cdk1-cyclinB is only regulated by the rate of cyclinB degradation. However, post-translational modifications (namely phosphorylations /dephosphorylations) are known to be important in regulating Cdk1-cyclinB activity. Could these regulations play a role in the local activation or inhibition of Cdk1-cyclinB activity to initiate the SCW, and could the authors implement in their reaction-diffusion model such events?

The Reviewer raises a point which we are very much looking forward to exploring in the future. Specifically, we are very interested in studying the precise molecular details shaping cdk1-cyclinB activity gradients in starfish oocytes and other systems. However, in this current study we wanted to illustrate that it is possible for a cdk1-cyclinB gradient to be built up by using the simplest model with minimal assumptions, a small set of parameters, and realistic parameter values. Additionally, we aimed to use assumptions that are common between published models of the cdk1-cyclinB system (Ciliberto et al. 2005; Yang & Ferrell 2013). It was great to see that such a minimalistic model resulted in a gradient that in its properties matched closely our experimental observations. We leave it to future work to develop a more detailed model and to parametrize and validate it against experiments. We have however expanded the text in the results section to make the goal of our model clearer (see line 163).

6 - On the same line, to further prove that the degradation of cyclinB is key to generate the cdk1-cyclinB gradient, could the authors express non-degradable cyclinB in their oocytes (to test if it suppressed the gradient and SCW) and/or modulate SCW speed by overexpressing WT cyclinB at different concentrations (as in Fig S3a, but with different concentrations, a way to modulate the gradient in time and slow down SCW in a dose dependent manner)?

We tried the experiment proposed by the Reviewer, and we could confirm the observations of Bement et al. (Bement et al. 2015) that the cell cycle can be halted and thus the SCW prevented by injecting non-degradable cyclinB. However, we were not able to titrate down the protein concentration to achieve intermediate effects likely due to the non-linearity of the response. As an alternative approach, we applied the proteasome inhibitor MG-132, which at the right doses only slowed down anaphase, rather than inhibiting it completely. For these intermediate MG-132 doses we observed a gradual increase of the time between NEBD and anaphase, consistent with a slowed degradation of cyclinB. Concomitant with this cell cycle delay we observed reduced contraction strength, which is consistent with the predictions of our model that increased time in metaphase will allow a further smoothing out of the gradient. Unfortunately, MG-132 caused apparent side effects resulting in abnormal shape changes in addition to SCWs, which overall rendered these experiments difficult to interpret. Therefore, we show these data here (Letter Fig. 5), but despite a large number of trials, we could not gain sufficient confidence in these experiments to show them in the manuscript.

To the second sub-question and as already hinted at above, in reality there are likely additional layers of regulation that render the response of the system more complex and non-linear. Correspondingly, the ‘dose-response’ experiments with cyclinB levels are non-trivial as they are very sensitive to the concentration of the cyclinB protein, which we are overexpressing over a background of endogenous cyclinB of unknown concentration and comparability between cells. Therefore, while performing a precise dose-response experiment would be very interesting it is practically not feasible in the current context.

References

- Abe, Y. et al., 2010. A single starfish Aurora kinase performs the combined functions of Aurora-A and Aurora-B in human cells. *Journal of cell science*, 123(Pt 22), pp.3978–88. Available at: <http://jcs.biologists.org/cgi/doi/10.1242/jcs.076315> [Accessed May 22, 2017].
- Bement, W.M. et al., 2015. Activator–inhibitor coupling between Rho signalling and actin assembly makes the cell cortex an excitable medium. *Nature Cell Biology*, 17(11), pp.1471–1483. Available at: <http://www.ncbi.nlm.nih.gov/pubmed/26479320> [Accessed December 13, 2016].
- Besser, A. & Schwarz, U.S., 2007. Coupling biochemistry and mechanics in cell adhesion: a model for inhomogeneous stress fiber contraction. *New Journal of Physics*, 9(11), pp.425–425. Available at: <http://arxiv.org/abs/0707.2551> [Accessed January 13, 2017].
- Borrego-Pinto, J. et al., 2016. Distinct mechanisms eliminate mother and daughter centrioles in meiosis of starfish oocytes. *The Journal of Cell Biology*, 212(7), pp.815–827. Available at: <http://www.jcb.org/lookup/doi/10.1083/jcb.201510083> [Accessed May 22, 2017].
- Chang, J.B. & Ferrell, J.E., 2013. Mitotic trigger waves and the spatial coordination of the Xenopus cell cycle. *Nature*, 500(7464), pp.603–607. Available at: <http://www.ncbi.nlm.nih.gov/pubmed/23863935> [Accessed August 6, 2013].
- Ciliberto, A. et al., 2005. Rewiring the exit from mitosis. *Cell cycle (Georgetown, Tex.)*, 4(8), pp.1107–12. Available at: <http://www.ncbi.nlm.nih.gov/pubmed/15970669> [Accessed December 19, 2016].
- Endo, T. et al., 1987. Structures of the sugar chains of a major glycoprotein present in the egg jelly coat of a starfish, *Asterias amurensis*. *Archives of biochemistry and biophysics*, 252(1), pp.105–12. Available at: <http://www.ncbi.nlm.nih.gov/pubmed/3813529> [Accessed May 22, 2017].
- Goryachev, A.B. et al., 2016. How to make a static cytokinetic furrow out of traveling excitable waves. *Small GTPases*, 1248(April), pp.1–6. Available at: <http://www.tandfonline.com/doi/full/10.1080/21541248.2016.1168505>.

- Hamaguchi, M.S. & Hiramoto, Y., 1978. Protoplasmic movement during polar-body formation in starfish oocytes. *Experimental cell research*, 112(1), pp.55–62.
- Hamaguchi, Y., Numata, T. & K Satoh, S., 2007. Quantitative analysis of cortical actin filaments during polar body formation in starfish oocytes. *Cell structure and function*, 32(1), pp.29–40. Available at: <http://www.ncbi.nlm.nih.gov/pubmed/17575411> [Accessed August 21, 2016].
- Mammoto, A. et al., 2009. A mechanosensitive transcriptional mechanism that controls angiogenesis. *Nature*, 457(7233), pp.1103–1108. Available at: <http://www.nature.com/doi/10.1038/nature07765> [Accessed February 2, 2017].
- Metz, C., 1985. *Biology Of Fertilization V1 : Model Systems And Oogenesis.*, Elsevier Science. Available at: <https://books.google.de/books?id=m5clgDAu6jkC&pg=PA139&lpg=PA139&dq=schoenmaker+starfish+oocyte&source=bl&ots=CidF87a9of&sig=vNFRUZb6h2SxNu354lsjOSb7no4&hl=en&sa=X&ved=0ahUKEwjNsaDTzYPUAhWM7hoKHf6aDclQ6AEIjzAA#v=onepage&q=schoenmaker+starfish+oocyte&f=false> [Accessed May 22, 2017].
- Moorhouse, K.S. et al., 2015. Influence of cell polarity on early development of the sea urchin embryo. *Developmental Dynamics*, 244(12), pp.1469–1484. Available at: <http://www.ncbi.nlm.nih.gov/pubmed/26293695> [Accessed May 22, 2017].
- Priya, R. et al., 2015. Feedback regulation through myosin II confers robustness on RhoA signalling at E-cadherin junctions. *Nature Cell Biology*, 17(10), pp.1282–1293. Available at: <http://www.ncbi.nlm.nih.gov/pubmed/26368311> [Accessed December 14, 2016].
- Robin, F.B. et al., 2016. Excitable RhoA dynamics drive pulsed contractions in the early *C. elegans* embryo. *bioRxiv*. Available at: <http://biorxiv.org/content/early/2016/09/21/076356> [Accessed June 5, 2017].
- Rosenberg, M.P., Hoesch, R. & Lee, H.H., 1977. The relationship between 1-methyladenine induced surface changes and fertilization in starfish oocytes. *Experimental cell research*, 107(2), pp.239–45. Available at: <http://www.ncbi.nlm.nih.gov/pubmed/577482> [Accessed May 22, 2017].
- Schroeder, P.C., Larsen, J.H. & Waldo, A.E., 1979. Oocyte-follicle cell relationships in a starfish. *Cell and tissue research*, 203(2), pp.249–56. Available at: <http://www.ncbi.nlm.nih.gov/pubmed/574803> [Accessed May 22, 2017].
- Terasaki, M. et al., 2001. A new model for nuclear envelope breakdown. *Molecular biology of the cell*, 12(2), pp.503–10. Available at: <http://www.ncbi.nlm.nih.gov/pubmed/11179431> [Accessed December 19, 2016].
- Terasaki, M. et al., 2003. Localization and Dynamics of Cdc2-Cyclin B during Meiotic Reinitiation in Starfish Oocytes. *Molecular biology of the cell*, 14(November), pp.4685–4694.
- Yang, Q. & Ferrell, J.E., 2013. The Cdk1–APC/C cell cycle oscillator circuit functions as a time-delayed, ultrasensitive switch. *Nature Cell Biology*, 15(5), pp.519–525. Available at: <http://www.ncbi.nlm.nih.gov/pubmed/23624406> [Accessed December 19, 2016].

Reviewers' Comments:

Reviewer #1:

Remarks to the Author:

The authors have thoughtfully and completely addressed the points I raised in my original review.

Reviewer #2:

Remarks to the Author:

The paper by Bischof et al. had improved after the revision. Generally, I am satisfied with the responses of the authors and their changes to the paper. Thus, the Discussion has been changed significantly and to the better.

Remaining issues that need to be taken care of:

In my question 3 I was asking how the model wave kymograph had been generated given that the authors did not simulate the full dynamic propagation model. While I understand that they stack together multiple static equilibria obtained for consecutive position of the band, I am not sure that the general reader, especially biologist will be able to understand this just looking at the Figure 1 and the caption to it. I require that the caption to Fig. 1d should be expanded to explain how such a kymograph was generated.

On a related subject, the authors have now added a sentence to the discussion: "Our observations that the speed of the SCW is not constant..." This sentence is surprising given that on page 4 they present SCW as a constant wave with velocity 42 ± 8 $\mu\text{m}/\text{min}$. Also, it is not clear how non-constant velocity is consistent with their kymographs on Fig. 1d. This is very confusing and needs to be clarified or corrected.

Technical issues associated with modeling:

Hamiltonian: 1) what is the definition of " l " variable? Angle? Arc length?

2) how is $\sigma(l)$ is defined? Gaussian? If something else, the authors have to specify how they determine the width of this function.

Biochemical model of Cdk1/APC interaction:

1) what is the initial concentration of APC/C? This should be entered into Table 1.

2) The Cdk1-inhibitable degradation of Cdk by APC term ($\text{APC}/(J + \text{cdk1}^2)$) has a dimensionality problem. It appears the authors forgot a constant with dimensionality $1/\text{concentration}$ in front of the cdk squared term.

3) What is the Theta capital function? Is this Heaviside function? Please define.

Reviewer #3:

Remarks to the Author:

The points raised in the previous round of review have been satisfactorily addressed, in fact beyond my expectations. Some comments I had could have been answered without performing additional experiments, and some experiments I asked for were really challenging. However, the authors did their best to answer to ALL my comments, adding new data and convincing explanations in the text where needed. As I said in my previous review, this is a high quality research. The findings are novel and very interesting. I am looking forward to their follow up studies.

Point-by-point response to the reviewers

Reviewer #1 (Remarks to the Author)

The authors have thoughtfully and completely addressed the points I raised in my original review.

We thank the reviewer for their comments and time in reviewing our manuscript and are glad that we could address all points raised.

Reviewer #2 (Remarks to the Author):

The paper by Bischof et al. had improved after the revision. Generally, I am satisfied with the responses of the authors and their changes to the paper. Thus, the Discussion has been changed significantly and to the better.

We would like to thank the reviewer for their detailed comments, which helped to significantly improve the manuscript, the Discussion in particular.

Remaining issues that need to be taken care of:

In my question 3 I was asking how the model wave kymograph had been generated given that the authors did not simulate the full dynamic propagation model. While I understand that they stack together multiple static equilibria obtained for consecutive position of the band, I am not sure that the general reader, especially biologist will be able to understand this just looking at the Figure 1 and the caption to it. I require that the caption to Fig. 1d should be expanded to explain how such a kymograph was generated.

We thank the reviewer for pointing out this issue. We updated the legend of Figure 1 to include a brief description of how the kymograph was generated (line 507).

On a related subject, the authors have now added a sentence to the discussion: "Our observations that the speed of the SCW is not constant..." This sentence is surprising given that on page 4 they present SCW as a constant wave with velocity 42 ± 8 $\mu\text{m}/\text{min}$. Also, it is not clear how non-constant velocity is consistent with their kymographs on Fig. 1d. This is very confusing and needs to be clarified or corrected.

We thank the reviewer for pointing out that this wording might be confusing. In untreated oocytes the speed of the SCW is indeed within a reasonably narrow range, although there is a significant variability as indicated by the standard deviation – likely to stem from the natural variability in the shape of the oocytes. Importantly, this variability in SCW speed was dramatically increased when we artificially manipulated the shape of the oocytes in microfabricated chambers. In these manipulated oocytes we could show that the speed correlates with the distance between the animal and vegetal pole. This indicates that the cdk1-cyclinB gradient is scalable and it also excludes the possibility of the SCW wave front being guided by a trigger wave of cdk1 inactivation (in which case we would expect the speed to be constant in all cases and independent of cell shape).

We now changed the phrasing in the Discussion to make it clear that we are referring there to the oocytes of manipulated shapes (line 215).

Technical issues associated with modeling:

Hamiltonian: 1) what is the definition of "l" variable? Angle? Arc length?

We appreciate the chance to further clarify details of our model. The variable 'l' does indeed denote arc length, which is the only relevant spatial coordinate as we assume rotational symmetry. We now include this detail in the Methods section (line 326).

2) how is $\sigma(l)$ is defined? Gaussian? If something else, the authors have to specify how they determine the width of this function.

The function $\sigma(l)$ is taken to be a half period of a sine. We chose this shape over a Gaussian because it is more localized than a Gaussian. With the Gaussian, it shares the features that it is

symmetric around the maximum and decays to the sides. The width is determined by the end to end distance, where the function goes to zero. We now clarify this in the Methods section (line 321).

Biochemical model of Cdk1/APC interaction:

1) what is the initial concentration of APC/C? This should be entered into Table 1.

The initial concentration of APC/C is $2500 \mu\text{m}^{-3} = 4 \mu\text{M}$ in the nucleus and $500 \mu\text{m}^{-3} = 0.8 \mu\text{M}$ in the cytosol. This data is included in Supplementary Table 2, alongside all the variables for the Cdk1/APC model.

2) The Cdk1-inhibitable degradation of Cdk by APC term ($\text{APC}/(J + \text{cdk1}^2)$) has a dimensionality problem. It appears the authors forgot a constant with dimensionality 1/concentration in front of the cdk squared term.

We thank the reviewer for this question as we had indeed here lost a squared term. We have corrected this mistake (Line 347). The term now correctly reads: $\frac{\text{APC}}{J^2 + \text{cdk1}^2}$.

3) What is the Theta capital function? Is this Heaviside function? Please define.

It is indeed the Heaviside function and we have now included this in the description (Line 349).

Reviewer #3 (Remarks to the Author)

The points raised in the previous round of review have been satisfactorily addressed, in fact beyond my expectations. Some comments I had could have been answered without performing additional experiments, and some experiments I asked for were really challenging. However, the authors did their best to answer to ALL my comments, adding new data and convincing explanations in the text where needed. As I said in my previous review, this is a high quality research. The findings are novel and very interesting. I am looking forward to their follow up studies.

We thank the reviewer for their very positive comments and are glad that our follow-up experiments and explanations included in the revised version could answer all their questions.